# Break it, Imitate it, Fix it:
# Robustness by Generating Human-Like Attacks

**Aradhana Sinha**[*1], **Ananth Balashankar**[*1], **Ahmad Beirami**[1],
**Thi Avrahami**[1], **Jilin Chen**[1], **and Alex Beutel**[†2]

[1]Google Research
[2]OpenAI

**Reviewed on OpenReview:** https://openreview.net/forum?id=UAT4j3Y7HP

## Abstract

Real-world natural language processing systems need to be robust to human adversaries. Collecting examples of human adversaries for training is an effective but expensive solution. On the other hand, training on synthetic attacks with small perturbations—such as word-substitution—does not actually improve robustness to human adversaries. In this paper, we propose an adversarial training framework that uses limited human adversarial examples to generate more useful adversarial examples at scale. We demonstrate the advantages of this system on the ANLI and hate speech detection benchmark datasets—both collected via an iterative, adversarial human-and-model-in-the-loop procedure. Compared to training only on observed human attacks, also training on our synthetic adversarial examples improves model robustness to future rounds. In ANLI, we see accuracy gains on the current set of attacks ($44.1\% \rightarrow 50.1\%$) and on two future unseen rounds of human generated attacks ($32.5\% \rightarrow 43.4\%$, and $29.4\% \rightarrow 40.2\%$). In hate speech detection, we see AUC gains on current attacks ($0.76 \rightarrow 0.84$) and a future round ($0.77 \rightarrow 0.79$). Attacks from methods that do not learn the distribution of existing human adversaries, meanwhile, degrade robustness.

## 1 Introduction

Robustness to real human-generated adversarial examples is crucial to reliable natural language processing (NLP) systems. While training on past real attacks improves robustness to future real attacks (Kiela et al., 2021), gathering such data is costly and time-consuming (Xu et al., 2021; Hendrycks et al., 2021). This motivates us to maximize model robustness using a fixed dataset of human-generated adversarial examples, without the need for additional human effort or larger better model architectures. We achieve this goal by leveraging synthetic examples we craft to mimic real attacks. This approach enables more cost-effective and scalable NLP security.

Adversarial examples are designed to induce misclassification (Szegedy et al., 2014). Adversarial robustness is the ability of a classifier to correctly label adversarial examples. Though adversarial robustness has been extensively studied on benchmark NLP tasks (Jia & Liang, 2017; Ettinger et al., 2017; Zhang et al., 2020; Gao et al., 2018), NLP classifiers often falter against real-world text adversaries (Lees et al., 2021; Borkan et al., 2019) due to the limitations of current solutions. These failures can result in real-world harms (Scheuerman et al., 2021).

Current NLP robustness solutions typically fall into two categories: training on human-generated (Dinan et al., 2019; Nie et al., 2020) or on synthetically generated adversarial examples (Uesato et al., 2018; Jin et al., 2020). There are not enough human-generated datasets as these are expensive to create (Xu et al., 2021; Hendrycks et al., 2021) — only becoming even more prohibitively so as models get larger and have more use cases (Ganguli et al., 2022). Synthetic attacks, meanwhile, (Uesato et al., 2018; Jin et al., 2020)

---

[*]Equal contribution.
[†]Work done while author was at Google.

often oversimplify adversarial attacks. Unlike in computer vision, small changes in text (discrete tokens) can drastically alter meaning, making it hard to craft synthetic examples with the desired label. This leads to oversimplified attacks based on easy metrics, which may not translate to robustness against real humans (Uesato et al., 2018).

Popular synthetic NLP attacks include (a) template-based or small text-edit-distance attacks, (Malfa & Kwiatkowska, 2022; McCoy et al., 2019b; Zang et al., 2020; Ren et al., 2019), (b) perturbation attacks that use word embeddings and search within an $\varepsilon$-neighborhood (Jia et al., 2019; Zhao et al., 2017; 2018; Li et al., 2021; Huber et al., 2022), or (c) finding universal adversarial perturbations (Moosavi-Dezfooli et al., 2017; Wallace et al., 2019; Mehrabi et al., 2022). Real attackers, meanwhile, are (i) known to make much larger edits from the original text, and (ii) are informed by each other's successful attacks, neither of which is captured in existing synthetic NLP attacks (West, 2017). In this paper, we take a step towards closing this gap by directly modeling the real attack patterns. This enables us to emulate human text attacks more realistically by (i) allowing larger edits and (ii) using existing real attacks to inform future attacks.

In prior work, the following proxies are usually used to measure whether generated adversarial examples are of good quality: semantic proximity (Malfa & Kwiatkowska, 2022), high attack success rate (Szegedy et al., 2014), low label noise (Malfa & Kwiatkowska, 2022), or distributional similarity to past attacks (Pillutla et al., 2021). These metrics, however, do not connect well to the attack patterns used by real adversaries. Our primary metric for attack quality is whether the generated attacks are useful when used in adversarial training to defend against future unseen rounds of human-generated attacks. That is, can we increase robustness beyond what we achieve by only training on all existing observed human attacks? We leverage frameworks like Dynabench to test on the evolving patterns of real adversaries (Kiela et al., 2021).

We show that our attack generation methods, which learn the distribution of human adversarial examples, outperform both (1) attack generators that do not take the human examples into account, and (2) attack generators that do not learn the distribution but rely on random perturbations (Sec 6). Our attack generation methods are able to make improvements even when trained on as few as 500 real human adversarial examples. Finally, though prior adversarial literature places a high emphasis on adversary success rates, low label noise, or distributional similarity, we show that these quality proxies are not predictive of whether an attack generator can better defend against future attacks. Our primary contributions are to:

1. **Demonstrate misalignment between synthetic and real attacks:** We empirically show that existing synthetic attack approaches do not necessarily improve robustness to the real attacks from humans.

2. **Overcome misalignment by imitating real adversaries:** We use generative models to directly imitate existing human-generated attacks. Our metric of success is how much we can improve robustness to future real attacks (beyond what can be accomplished by adversarially training on all existing real attacks).

3. **Improve adversarial robustness without relying on a better/bigger model:** Adversarial training on imitated real attacks provides significant robustness benefits. When compared to solely training on existing real attacks, we improve accuracy by 11% on unseen attacks in the ANLI benchmark, and by 8% on existing attacks in the hate speech detection benchmark.

4. **Show misalignment between common attack quality metrics and attack usefulness in preventing future attacks:** We empirically show that more distributional similarity, low label noise, or high adversary success rate do not entail that an attack generator is better than another in helping defend against downstream attacks.

## 2 Related Work

**Adversarial robustness** is measured as accuracy on challenge sets such as Adversarial GLUE (Wang et al., 2021), which are gathered through crowd-sourcing and programmatic text perturbation (Zhang et al., 2019; Morris et al., 2020b). To improve adversarial robustness, prior work uses training interventions and/or data augmentation. Training interventions focus on learning more meaningful stable representations from

available data: e.g. using mutual information (Wang et al., 2020a; Zhao et al., 2022) or locally regularizing (Aghajanyan et al., 2020; Zhu et al., 2019). In contrast, we focus on data augmentation, which may be used with these training intervention methods.

Despite the popularity of data augmentation solutions such as $\epsilon$-bounded PGD attacks (Madry et al., 2017) in continuous domains, it is not straightforward to extend them to discrete NLP settings. In NLP, small perturbations to the text can have a big impact on the text's true label.

Nevertheless, **controlled text generation** has vastly improved in recent years through zero or few-shot tuning of large language models (Wu et al., 2021; Perez et al., 2022). Such methods use data augmentation for fine-tuning (Michel et al., 2019b; Garg & Ramakrishnan, 2020), gradients from small attribute classifiers to ensure generated text has a particular class (Dathathri et al., 2020; Wu et al., 2023), or careful prompting to generate semantically close text with only the desired attribute swapped (Madaan et al., 2020).

*Adversarial* controlled text generation, however, is is even more challenging. It's not straightforward to correctly label the generated text we intend to use as an adversarial example (by definition). Prior work typically gets around this challenge by making very small or rule-based perturbations that should not change the original example's true label (They assume low label noise is a good criteria for an attack generation method). These include contextual synonym-substitution (Michel et al., 2019a; Jia et al., 2019; Li et al., 2020a; Morris et al., 2020a), rule-based grammar (McCoy et al., 2019a; Ribeiro et al., 2018), morphological (Tan et al., 2020), and character-level (Eger & Benz, 2020) manipulations. A related but under-explored area is to adversarially intervene on all parts of the text that are invariant to the predicted label (Wang et al., 2020b; Chen et al., 2021; Lei et al., 2022), or conversely to minimally intervene to ensure the true label changes (Ross et al., 2021; Deng et al., 2022). Our attack method is different in that it relies on memorizing and re-mixing phrases from existing adversarial examples to generate additional adversarial examples that the attack method can correctly label. All of these methods, when used for adversarial training, improve robustness to their respective attack type. In this paper, however, we ask whether they improve robustness against human adversaries.

Most of prior work assumes that adversarial text should be fluent, grammatically correct or semantically similar to the original. There is no such constraint on adversaries in the real world. Hence, in this paper we use the Hate Detection task which often does not have fluent grammatical text. Prior work that does not assume a standardized single language, include studies on dialectal language like Ebonics on Twitter (Blodgett et al., 2016), multilingual text on Reddit (Michel & Neubig, 2018), Hinglish (Biradar et al., 2021)), emoji-based hatespeech (Kirk et al., 2022); and methods that seek to distinguish human and machine generated text through heuristics (e.g.: sentence length, verb ratios, (Yao et al., 2017), relational consistency (Zhong et al., 2020)).

**Red teaming:** We are also motivated by red-team human generated datasets—human-created attacks that target a specific model, often by crowd-sourcing. Examples include SWAG, ReCoRD, HotpotQA, HellaSWAG, HANS (Zellers et al., 2018; Zhang et al., 2018; Yang et al., 2018; Zellers et al., 2019; McCoy et al., 2019a; Kaushik et al., 2019). Sometimes the work also uses feedback from the original classifier: CoDAH, Quoref, DROP, FEVER 2, ANLI, etc. (Chen et al., 2019; Dasigi et al., 2019; Dua et al., 2019; Thorne et al., 2019; Nie et al., 2020; Bartolo et al., 2020; Kiela et al., 2021). We rely on such human-model interactions in a feedback loop set-up, and propose a mechanism to reduce red team costs.

## 3 Problem Formulation

Red-teaming is popular because training on past real attacks improves robustness to future real attacks (as depicted in Fig.1(a)). Real attacks, however, are costly to source. Thus for a fixed dataset of past human attacks, our goal is to improve robustness to future attacks even further — without resorting to sourcing more human attacks (and without resorting to a larger or more capable model architecture). We achieve this goal by making use of synthetically generated examples as depicted in Fig. 1(b).

We evaluate robustness – our success criteria– as performance on successive rounds of adversarial examples collected through the crowd-sourced model-in-the-loop approach (Nie et al., 2020) in Fig. 1(a). We take the

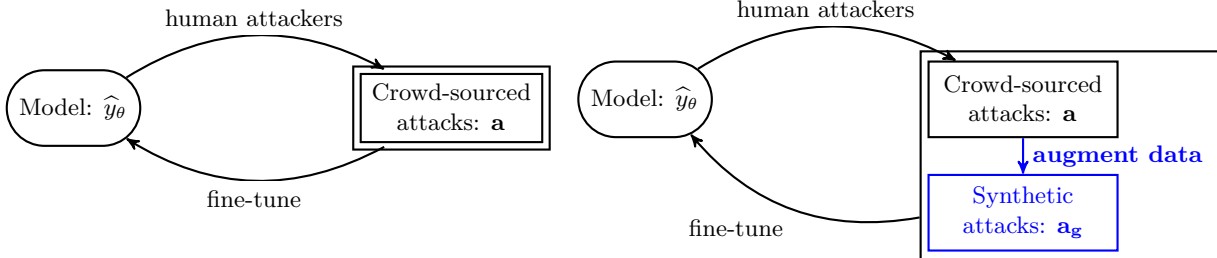

(a) Human in the Loop Adversarial Training

(b) Human in the Loop Adversarial Training With Synthetic Data Augmentation

Figure 1: In traditional human-in-the-loop adversarial training, humans attack attack a model, and then the model learns from those attacks to become more robust (Fig. 1(a)). Augmenting the human-generated attacks with synthetic attacks is a popular way to increase robustness (Min et al., 2020) (Fig. 1(b)). We propose two new methods that generate synthetic adversarial attacks by learning the patterns of real crowd-sourced attacks. Our methods significantly outperform existing techniques in defending against future yet-unseen crowd-sourced attacks. Such prior work on synthetic attacks does not typically learn patterns from real crowd-sourced attacks as we do; they focus on making small edits that make the attack harder while ensuring low label noise (Feng et al., 2021).

classifier architecture to be fixed; improvements to architecture or model size are out of scope. We focus only on the data augmentation method.

**Notation: (See Fig. 1)** Let $\mathbf{x}$ be a collection of samples drawn from some distribution $\mathbf{X} : \mathbf{x} \sim \mathbf{X}$, with ground truth labels $y(\mathbf{x})$. There is a classifier, parameterized by $\theta_0$, that is trained on $\mathbf{x}$. This classifier outputs label predictions: $\widehat{y}_{\theta_0}(\mathbf{x})$. Human adversaries attack classifier $\theta_0$ to create a dataset of crowd-sourced attacks, $\mathbf{a}_0 \sim \mathbf{X}_{\mathbf{a}_0}$. All examples in $\mathbf{X}_{\mathbf{a}_0}$ fool the classifier, i.e., $\widehat{y}_{\theta_0}(\mathbf{a}) \neq y(\mathbf{a}) \forall \mathbf{a} \in \mathbf{X}_{\mathbf{a}_0}$.

The classifier is then further fine-tuned on this dataset of real attacks, $\mathbf{a}_0$. We analogously refer to the predictions of this newly fine-tuned classifier as $\widehat{y}_{\theta_1}(\mathbf{x})$. Again, human adversaries attack classifier $\theta_1$ to create a dataset of crowd-sourced attacks, $\mathbf{a}_1 \sim \mathbf{X}_{\mathbf{a}_1}$. All examples in $\mathbf{X}_{\mathbf{a}_1}$ fool the classifier, i.e., $\widehat{y}_{\theta_1}(\mathbf{a}) \neq y(\mathbf{a}), \forall \mathbf{a} \in \mathbf{X}_{\mathbf{a}_1}$.

This model-in-the-loop process results in evolving crowd-sourced real attacks and evolving predictions as follows:

$$\text{Model: } \widehat{y}_{\theta_0} \rightarrow \text{Crowd-sourced attacks: } \mathbf{a}_0 \rightarrow \text{Model: } \widehat{y}_{\theta_1} \rightarrow \text{Crowd-sourced attacks: } \mathbf{a}_1 \rightarrow \cdots \quad (1)$$

**Our goal is to improve the accuracy of $\widehat{y}_{\theta_1}$ on these yet unseen future attacks: $\mathbf{a}_1$, $\mathbf{a}_2$, without 1) gathering additional human attacks beyond $\mathbf{a}_0$ or 2) resorting to using a larger or better model architecture for $\widehat{y}_{\theta_1}$.** We accomplish this by also fine-tuning on synthetic data (See Fig. 1(b)) as follows.

$$\text{Model: } \widehat{y}_{\theta_0} \rightarrow \text{Crowd-sourced attacks: } \mathbf{a}_0, \underline{\text{Synthetic attacks: } \mathbf{a_g}} \rightarrow \text{Model: } \widehat{y}_{\theta_1} \quad (2)$$

Note we do not have access to $\mathbf{X}_{\mathbf{a}_1}, \mathbf{X}_{\mathbf{a}_2}$ or $\mathbf{a}_1, \mathbf{a}_2$ while training $\widehat{y}_{\theta_1}$. As such, we do not have guarantees on whether $\mathbf{a}_1, \mathbf{a}_2$ will be similar to $\mathbf{a}_0$, but we do know they are generated using the same crowd-sourcing methods.

## 4   Methods

To address the problem in Sec. 3, we describe how we use synthetic attacks to improve robustness. Then we describe the specific methods of generating the synthetic attacks.

**Overall Solution Framework**

As depicted in Fig. 1(b), we aim to be more robust to unseen future human attacks $(\mathbf{a}_1, \mathbf{a}_2)$ by fine-tuning our classifier $\widehat{y}_{\theta_0}$ not only on past observed attacks $(\mathbf{a}_0)$, but also on additional synthetic examples $(\mathbf{a_g})$. We create these generated examples with a generator, $G$. This generator learns what existing adversarial examples, $\mathbf{a}_0 \sim \mathbf{X}_{\mathbf{a}_0}$ looks like, and uses this learned approximation to generate synthetic examples: $\mathbf{a_g} \sim \widehat{\mathbf{X}_{\mathbf{a}_0}}$. Notably, we ensure the model we use for $G$ is no larger and no more capable at the target task than the original classifier $\widehat{y}_{\theta_0}$, that was fooled by the past observed adversarial examples $(\mathbf{a}_0)$. We want the improvements to come from the data augmentation method, not from the knowledge of a better model. In our experiments, $G$ is fine-tuned T5 model, and the classifier is fine-tuned Bert-Large.

We have no guarantees that future unseen attacks $(\mathbf{a}_1, \mathbf{a}_2)$ will be similar to the past seen attacks $(\mathbf{a}_0)$. We know, however, that they share an attack generation process. Thus, the hypothesis of this work is that modeling the existing real attack distribution allows the generator $(G)$ to capture something about the real attack generation process more broadly. We hypothesize the generator can generalize to future unseen attacks. **Restated, we hypothesize training on $\widehat{\mathbf{X}_{\mathbf{a}_0}}$ will not only make the classifier robust to $\mathbf{X}_{\mathbf{a}_0}$, but also reasonably robust to future adversarial attack distributions, $\mathbf{X}_{\mathbf{a}_1}$, $\mathbf{X}_{\mathbf{a}_2}$.**

A caveat: while in practice $\mathbf{a}_1$ and all subsequent attacks would depend on what can fool the new model $\theta_1$ created by training on the synthetic examples $(\mathbf{a_g})$ also (See Fig. 1(b)), we do not generate new human attacks in response to the new classifiers we train. Hence $\mathbf{a}_1$ and $\mathbf{a}_2$ are fixed in our setting from the original set-up in Fig. 1(a). The new rounds of attack are often also generated by fooling larger more capable models we do not use (ex: RoBERTa Large instead of BERT Large) (Kiela et al., 2021). Instead we show that future human attacks $(\mathbf{a}_1,\mathbf{a}_2)$ are less effective thanks to training on $\mathbf{a_g}$, than they would be by only traning on past observed examples $\mathbf{a}_0$.

**Synthetic Attack Generators**

For many NLP tasks, the input $x$ can be broken into $(x_i, x_o)$. $x_i$ is the portion of the input text that is not attacked by the adversary. It remains the same. This can be context: e.g. premise in NLI, paragraph in QA tasks, etc. For the toxicity task, where the entire sentence may be attacked, we set $x_i$ to be the first half of the text. $x_o$ is the portion that is attacked: e.g. hypothesis in NLI, question in QA, comment in sentiment analysis. We now present two methods to generate $\mathbf{a_g} \sim \widehat{\mathbf{X}_{\mathbf{a}_0}}$. The first is a imitation-only approach agnostic to the task. The second takes the classifier task into account to better maintain the desired class label.

**Method 1. Direct Imitation (DI): Label-aware fine-tuning** We fine-tune the generative model on existing observed attacks, $\mathbf{a}_0 \sim \mathbf{X}_{\mathbf{a}_0}$. We consider $(x_i, y)$ as the input for the generator, and $x_o$ to be the target text that needs to be generated. Incorporating $y$ as input greatly helps reduce the rate of noisy labels— the rate at which the the generated example does not retain the same label as the input $(y(\mathbf{a_g}) \neq y(\mathbf{a}_0))$. Specifically, we minimize the cross-entropy loss of the generated text probabilities $\widehat{\mathbf{X}_{\mathbf{a}_0}}(x_i, y)$ and the target text $x_o$. See Appendix A for additional implementation details and loss function definition.

**Method 2. Imitation + Controlled Exploration (ICE):** The primary challenge of the DI method, and the primary challenge of all controlled adversarial text generation more broadly, are noisy labels (By definition, adversarial examples are hard to label correctly when we restrict ourselves from relying on a bigger better model).

To overcome this challenge, we modify the Plug and Play controlled decoding method to make it suitable for adversarial robustness (Dathathri et al., 2020). There are three key components:

1. A text generation transformer model, $T$, that can generate free-form text. It has been trained on a reconstruction loss: given $x_i$, it must output the corresponding $x_o$.

2. Attribute classifiers, $C$. This is a single layer feed-forward network with a cross-entropy loss — a very simple model that focuses specifically on predicting the task label, $y$. This classifier is used to guide the tokens picked by the text generation model, $T$.

| $\mathbf{a_0}$ | The Nassau County population increased from 2010 to 2016. |
|---|---|
| $\mathbf{a_0}$ | The Crystal Mountain Resort is a tourist destination. |
| $\mathbf{a_g}$ | Crystal Mountain population increased from 2010 to 2016. |

Table 1: An example of the ICE attack generator remixing existing observed attacks (top two) from the ANLI R1 data to create a new attack (bottom).

3. A steering signal. This is the information used by the attribute classifier $C$ to steer $T$. Our steering signal are the hidden layer activations after each transformer unit in $T$.

The system works as follows. We provide the generative model $T$ with a starting prompt $x_i$. $T$ generates some initial text based on it's understanding of what the matching $x_o$ is likely to be. The attribute classifier $C$ analyzes the text generated so far and assigns it a probability of belonging to the true label $y$ based on the steering signal. The generator $T$ slightly adjusts its steering signal to minimize the loss in $C$, and then picks a new token to generate. Thus the generator $T$ adapts its generated output with the desired true label. This process repeats for each token generated by $T$. Restated, the main generator $T$ generates some text similar to the original attack, while the attribute classifier $C$ nudges it to output text that has the correct label. This process is outlined in Algorithm 1.

**Warm-starting Trick:** This system only works well when the simple linear classifier $C$ makes reasonably correct predictions of the true label, $y$. Else it is unable to guide text generation effectively. We, however, are working with adversarial examples that are difficult even for a much larger model (including the base model of $T$) to correctly label. A simple linear classifier like $C$ is not likely to correctly label adversarial examples.

We mitigate this issue by training the system (both $T$ and $C$) on all past observed examples $\mathbf{a_0}$. And then passing the same examples $\mathbf{a_0}$ at inference time to generate synthetic examples.

Though this may run the risk of memorizing and replicating exactly the examples seen at training time, in practice however, this is not the case. There is diversity of synthetic text generated as each of the three components are imperfect. Further, we use a higher temperature to decode from $T$ at inference time, and because there are multiple $x_o$ associated with any given $x_i$ in our datasets, the adversarial examples generated by our ICE method do largely re-mix concepts from existing examples (See Table 1). For additional implementation details and loss function definitions, we refer to the Appendix A.

## 5  Experiments

This section details how we evaluate the methods listed in Section 4—the benchmark dataset used, implementation details, and baselines.

**A. Tasks: Human-in-the-loop crowd-sourced**

We want to assess whether we can meaningfully amplify past human-generated attacks to be more robust to future human-generated attacks (given fixed attack generation instructions and UI). Hence, we chose two very different DynaBench tasks: Natural Language Inference (NLI) and Hate Speech Detection: The former has longer and qualitatively more varied texts. The latter is terse, less varied, and has less standard English (often with incorrect grammar and spelling) (Kiela et al., 2021).

**A.I. Adversarial NLI:** We evaluate our methods on the Adversarial NLI (ANLI) task (Nie et al., 2020). This is a Natural Language Inference (NLI) task: the goal is to determine whether a *hypothesis* logically follows (entailment) or contradicts (contradiction) or is undetermined (neutral) based on facts present in the *premise*. Nie et. al. crowd-source human attacks on the hypothesis against a base classifier, $\widehat{y}_{\theta_0}$, trained on MNLI+SNLI data (Wang et al., 2018; Bowman et al., 2015). Then they train more robust models by incorporating these new attacks (and other data) for three rounds. $\mathbf{a_0}$ are the human generated attacks from the first round created by attacking $\widehat{y}_{\theta_0}$, a BERT-Large transformer classifier (Devlin et al., 2019).

---

**Algorithm 1** Pseudo-code for ICE Method

---

**Input:**
Adversarial dataset: $\forall (x, y) \in \mathbf{a}_0$, where $x = (x_i, x_o)$, $x_i$: unchanged context in text, $x_o$: text that is perturbed by the attacker; $y$: true label corresponding to $x$
$T$: a pre-trained encoder decoder transformer model that can do generation and classification
$S$: a vector of the activation states of the hidden layer after each transformer unit in $T$
$C$: a linear feed-forward classifier that takes $S(x)$ as input.

**Warm-start:**
Fine-tune $T$ on dataset $\mathbf{a}_0$ on the reconstruction task: $x_i \rightarrow x_o$
Fine-tune $T$ on dataset $\mathbf{a}_0$ on the classification task: $x \rightarrow y$
Freeze the parameters of $T$ and train $C$ on dataset $\mathbf{a}_0$ on the classification task: $S(x) \rightarrow y$

**Iterative Adapt and Generate:**
$T' \leftarrow$ copy of $T$
Freeze the parameters of $C$ and unfreeze that of $T$
**for all** $(x, y) \in \mathbf{a}_0$ **do**
    $a_g \leftarrow \emptyset$
    **while** end of sentence token $\notin a_g \wedge \text{length}(a_g) <$ maximum length **do**
        $a_g \leftarrow a_g + t$, $t$ : next token generated using the reconstruction task $T(x_i)$
        Backprop $\nabla_T \texttt{ClassifierTaskLoss}(C(S(x_i, a_g)), y) + \lambda \nabla_T \texttt{ReconstructionLoss}(a_g, x_o)$ for 10 steps
    **end while**
    **yield** $(x_i, a_g)$ : a synthetic adversarial example corresponding to $(x, y)$
    $T \leftarrow T'$
**end for**

---

Successive rounds $\mathbf{a}_1$, $\mathbf{a}_2$ are created by attacking RoBERTa models. We choose to improve the robustness of BERT-Large model $\widehat{y}_{\theta_0}$ using $\mathbf{a}_0$ and evaluate on future human adversarial attacks: $\mathbf{a}_1, \mathbf{a}_2$.

**A.II. Hate Speech Detection:** We also evaluate on the Dynabench Hate Speech detection dataset, an adversarial human-in-the-loop dataset generated in four rounds (Vidgen et al., 2021). In Round 1, Vidgen et. al. train a base RoBERTA classifier, $\widehat{y}_{\theta_0}$, on original content created by humans. In Rounds 2-4, they create more robust RoBERTa models ($\widehat{y}_{\theta_1}, \widehat{y}_{\theta_2}, \widehat{y}_{\theta_3}$) by training on attacks created as follows: Human raters first create original content that successfully fools the base classifier. Then they perturb these new sentences to create even more challenging "contrast sets" with different labels. This data is then split into train, validation, and test sets, with half the test set entries created by annotators who do not appear in the training and validation sets to minimize annotator bias.

Note that for both tasks we do not gather additional human adversarial attacks targeting our improved classifiers. We evaluate on a fixed set of attacks previously unseen by the classifier. This is a limitation of our set-up; gathering additional rounds of human attacks is left as future work.

## B. Base Attack Generator Model is T5

We use the T5 encoder-decoder as our generator for this paper (Raffel et al., 2020) for two reasons. First, it is compatible with multiple small sentence generation tasks (Raffel et al., 2020). Second, its performance on benchmark NLI tasks (MNLI, ANLI), and the Hate Speech task is close to that of the BERT-Large model we seek to improve – i.e. improvements are not coming from a superior model but rather imitating the real adversaries.

## C. Baselines

We have two types of baseline attack generators: The first type uses existing observed attacks, but does not learn the distribution, instead relying on random perturbations. We use three of these methods: TextFooler, BertAttack, and CT-GAN. TextFooler is a very popular attack generation library that transmutes the most

| Model | $a_0$: R1 | $a_1$: R2 | $a_2$: R3 |
|---|---|---|---|
| $\widehat{y}_{\theta_1}$: Base + R1 | $44.1_{\pm 0.03}$ | $32.5_{\pm 0.05}$ | $29.4_{\pm 0.05}$ |
| $\hookrightarrow$ + TextFooler(R1) | $24.1_{\pm 0.08}$ | $27.9_{\pm 0.06}$ | $30.3_{\pm 0.06}$ |
| $\hookrightarrow$ + BERT-Attack(R1) | $35.1_{\pm 0.12}$ | $29.0_{\pm 0.08}$ | $31.3_{\pm 0.09}$ |
| $\hookrightarrow$ + CT-GAN(R1) | $26.8_{\pm 0.14}$ | $29.5_{\pm 0.12}$ | $29.5_{\pm 0.11}$ |
| $\hookrightarrow$ + DI(MNLI+SNLI) | $22.9_{\pm 0.11}$ | $28.1_{\pm 0.12}$ | $29.4_{\pm 0.10}$ |
| $\hookrightarrow$ + ICE(MNLI+SNLI) | $33.9_{\pm 0.78}$ | $\underline{33.7}_{\pm 0.67}$ | $\underline{33.5}_{\pm 1.47}$ |
| $\hookrightarrow$ + DI(R1) | $\underline{48.2}_{\pm 0.32}$ | $\underline{39.1}_{\pm 0.29}$ | $\underline{\mathbf{40.2}}_{\pm 0.37}$ |
| $\hookrightarrow$ + ICE(R1) | $\underline{\mathbf{50.1}}_{\pm 1.43}$ | $\underline{\mathbf{43.4}}_{\pm 2.91}$ | $\underline{39.9}_{\pm 1.38}$ |

Table 2: Improvement on ANLI mean accuracy (%) (± standard error across 3 runs) when trained on attacks generated only from Round 1. The notation, DI(R1) for instance, refers to the method DI using R1 data to generate more examples. We underline the setups that outperform the *Base + R1* baseline.

| Model | $a_0$: R2 | $a_1$: R3 | $a_2$: R4 |
|---|---|---|---|
| Base + R1 + R2 | $0.76_{\pm 0.001}$ | $0.78_{\pm 0.003}$ | $0.77_{\pm 0.001}$ |
| $\hookrightarrow$ + TextFooler(R2) | $\underline{0.78}_{\pm 0.011}$ | $0.77_{\pm 0.012}$ | $0.76_{\pm 0.009}$ |
| $\hookrightarrow$ + BERT-Attack(R2) | $\underline{0.78}_{\pm 0.013}$ | $0.76_{\pm 0.015}$ | $0.77_{\pm 0.017}$ |
| $\hookrightarrow$ + DI(R2) | $\underline{\mathbf{0.84}}_{\pm 0.013}$ | $0.77_{\pm 0.034}$ | $0.76_{\pm 0.018}$ |
| $\hookrightarrow$ + ICE(R2) | $\underline{\mathbf{0.83}}_{\pm 0.032}$ | $0.80_{\pm 0.024}$ | $\underline{\mathbf{0.79}}_{\pm 0.018}$ |

Table 3: Improvement on Hate speech detection AUC (± standard error across 3 runs) when trained on attacks generated only from Round 2. The notation, DI(R2) for instance, refers to the method DI using R2 data to generate more examples. In this dataset, R1 is not adversarially generated, and is analogous to the base MNLI/SNLI data in the ANLI task.

predictive words, while preserving semantic similarity and contextual coherence (Jin et al., 2020). BertAttack is another popular method: it uses the model it's attacking to identify vulnerable words in the input; then, it uses BERT to generate substitutes for the vulnerable words. (Li et al., 2020b). CT-GAN is a Generative Adversarial Network(Goodfellow et al., 2014) modified for controlled text generation where the NLI premise is used as the control text (Haidar et al., 2019; Betti et al., 2020). The second type of baseline learns an example distribution, but does not use the attack distribution. We repeat our main methods, ICE and DI, in a new data setting. Instead of using ANLI R1, we use the MNLI+SNLI data used in to train the base classifier, $\widehat{y}_{\theta_0}$.

# 6 Results

**A. Synthetic human-like adversarial data improve robustness to future attacks. Distribution-agnostic baselines do not.**

Table 2 and Table 3 show that for both tasks, the accuracy on future rounds of human-generated attack ($a_1$, $a_2$) improves when generated examples, $a_g$ from ICE and DI are incorporated into training.

The adversarial example generators that attempt to imitate $a_1$ (ICE and DI) out-perform all types of baselines. First, they improve robustness beyond what we achieve by training on past human adversarial attacks, $a_1$, alone. This improvement cannot be achieved merely by training for more steps on ANLI R1 as shown in Table 18 in the Appendix. Second, they out-perform methods that rely on noise-based attacks on $a_1$ like TextFooler, BertAttack, and CT-GAN. Finally, they out-perform methods that imitate example distributions generated by other processes: ICE(MNLI+SNLI) and DI(MNLI+SNLI). The word/phrase substitution methods, BertAttack and TextFooler, improve accuracy within the same round for the Hate Speech dataset– this dataset is itself half-generated by making such minimal substitutions. Yet, these methods are not more effective on future rounds.

Table 17 extends the setting where R1 and R2 are past observed attacks available for training, and R3 is the held out set of future human attacks, leading to similar conclusions.

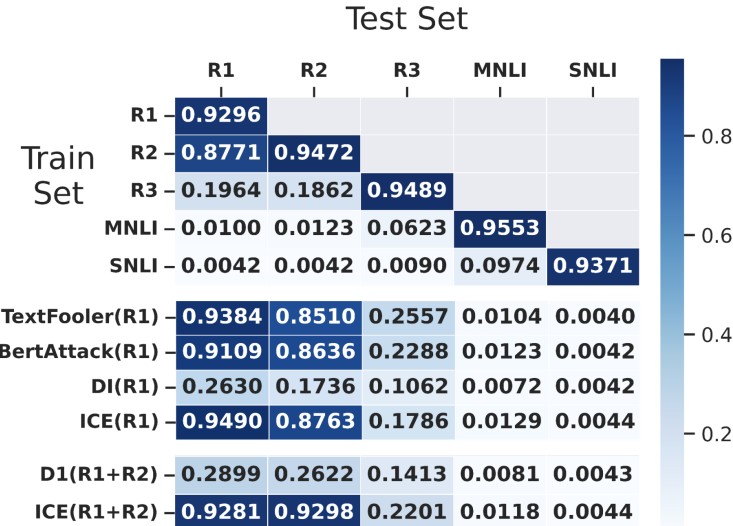

Figure 2: Distributional similarity, as measured by MAUVE on RoBERTa embeddings from a random 1k sample (Pillutla et al., 2021). MAUVE scores range from 0 to 1, with higher values indicating more similar distributions. MAUVE metrics are intended to be evaluated relative to each other, and not as absolute measures. Note that distributional similarity to the held out attacks, R2, does not correlate with whether an attack generation method is useful as per Table 2.

**B.  Common metrics for an attack method (label noise rate, attack success rate, and distributional similarity) do not entail whether adversarial data from the method can help defend against future attacks.**

Adversarial robustness literature considers attack methods to be better if they produce datasets with less label noise (Dathathri et al., 2020), or higher attack success rates (Uesato et al., 2018), or higher proximity to the original dataset (Ross et al., 2021). We show that these metrics are not good proxies for determining which type of attack method can generate examples that best defends against future attacks. These findings are surprising, and additional investigation is needed to understand why (Also see generated examples and additional correlation analyses in Appendix A).

**B.I.  Higher Distributional Similarity of $a_g$ to future attacks does not entail more useful adversarial examples.** We use MAUVE as a state-of-the-art metric of distributional similarity between two text datasets in Fig. 2 (Pillutla et al., 2021). We find that we cannot predict whether a synthetic dataset will improve the model's ability to resist future attack datasets simply by looking at how similar or dissimilar the two datasets are. For example, we trained the model on three synthetic datasets: DI(R1), ICE(R1), and TextFooler(R1). Though TextFooler(R1) is more distributionally similar to the R2 future attacks than DI(R1) (as per Fig. 2), DI(R1) is better at increasing robustness to R2 (as per Table 2). ICE(R1), meanwhile has both high distributional similarity and robustness. This finding is analogous to findings in computer vision: for example, training on $l_2$ Projected Gradient Descent attacks in MNIST increases robustness to Decision Boundary Attacks. These are two very different distributions of attacks: one created with gradient methods, and the other not (Madry et al., 2017; Brendel et al., 2018). A better understanding of how robustness transfers across different attack distributions is an open problem.

**B.II.  Lower label noise does not entail more useful adversarial examples.** One of the hardest challenges in controlled adversarial example generation is generating examples that have the desired label, that is reducing the rate of noisy labels. While this is a useful goal within an attack method type, comparing rates of noisy labels across attack methods does not help us choose a more useful method. Table 4 shows that TextFooler has more correct labels than the DI method, even though Table 2 makes it clear that DI is the more useful method. We, the authors, annotated 100 examples from each attack method and report the

| Generated attack dataset | Rate of correct labels |
|---|---|
| $\mathbf{a_g}$: TextFooler(R1) | 51% |
| $\mathbf{a_g}$: DI(R1) | 39% |
| $\mathbf{a_g}$: ICE(R1) | **78%** |

Table 4: Rate of correct labels from the attack generation methods on ANLI round 1 (↑ better). We, the authors, rated 100 random generated examples from each method to obtain these numbers. Our rating guidelines are included in the Appendix A, and the ratings are in Supplemental Materials.

| Model to attack → Attack Dataset ↓ | Base $\widehat{y}_{\theta_0}$ | Base+R1 $\widehat{y}_{\theta_1}$ | Base+R1+R2 $\widehat{y}_{\theta_2}$ |
|---|---|---|---|
| $\mathbf{a_0}$: R1 | 78% | 56% | 47% |
| $\mathbf{a_1}$: R2 | 73% | 67% | 58% |
| $\mathbf{a_2}$: R3 | 71% | 71% | 62% |
| $\mathbf{a_g}$: TextFooler(R1) | 96% | 98% | 94% |
| $\mathbf{a_g}$: DI(R1) | 76% | 87% | 79% |
| $\mathbf{a_g}$: ICE(R1) | 78% | 88% | 88% |

Table 5: Attack success rate of attack datasets on base BERT Large models as they incorporate more rounds of ANLI training data (↑ better). Only the examples that had the correct labels (as verified by human rating) are included in computing this rate.

rate of correct labels by comparing our annotated ground truth with the adversarial label associated with the corresponding (premise, hypothesis) pair in the ANLI R1 dataset. We refer to the Appendix A for tables of generated examples from each of the attack methods, and guidelines for human annotations.

**B.III. Higher adversary success rate does not entail more useful adversarial examples.** Table 5 shows the attack success rate of the various attack datasets (only including attacks with the correct label as verified by human ratings). All synthetic attack generators have high success rates all classifiers - which indicates that the information in the generated examples were not yet captured in by the base classifiers. Surprisingly, TextFooler, which has the highest attack success rate, does not improve adversarial robustness; R2, R3, DI(R1), and ICE(R1) datasets have lower attack success rates but are much more useful in increasing robustness to future attacks.

**C. Even ∼1k human adversarial examples improves robustness to unseen adversaries.**

We test how the number of human adversarial examples used to train the attack generator affects the adversarial robustness using the generated examples. We fix the number of examples we use to train $\widehat{y}_{\theta_1}$ at 10k generated examples. The full set of real adversarial examples from ANLI R1 is 16.9k real examples. We sample the number of real examples we use to train the attack generator. Table 6 demonstrates that increasing human examples improves robustness. Nevertheless, even when we provide low numbers of human examples, robustness to future rounds has improved. In-distribution accuracy on R1, however, suffers until 8k examples are used to train the attack generator. When we have fewer than 500 human adversarial examples to amplify using our approach, the robustness gains do not generalize. That happens because the few-shot setting creates several additional considerations that are out-of-scope in this paper:

- Simple fine-tuning baselines do not perform well in the few-shot setting. Other few-shot and parameter efficient baselines might be relevant (Zhou et al., 2022; Liu et al., 2022). The choice of training method is orthogonal to our goal of synthetic data generation.

- As we reduce the number of adversarial examples, evaluating generalization requires categorizing examples into various patterns to ensure coverage. In this work, we instead use the pattern-agnostic approach of Dynabench (Kiela et al., 2021), which aggregates all human-generated attacks.

.

| # R1 samples | $a_0$: R1 | $a_2$: R2 | $a_2$: R3 |
|---|---|---|---|
| 100 | 33.0 | 27.8 | 28.9 |
| 200 | 34.2 | 28.1 | 29.5 |
| 500 | 37.1 | 30.8 | 32.6 |
| 1024 | 42.7 | 36.5 | 36.1 |
| 2048 | 43.2 | 33.2 | 36.5 |
| 4096 | 43.3 | 37.8 | 37.1 |
| 8192 | 44.5 | 36.6 | 36.6 |
| all (16.9k) | **48.2** | **39.1** | **40.1** |

Table 6: Test accuracy (%) as we vary the number of R1 real adversarial examples used to fine-tune the DI attack generator. We use the trained DI generator to generate 10k examples. We fine-tune a Base + R1 classifier on these 10k DI(R1) examples to produce the robustness metrics above. We underline the setups that outperform the *Base + R1* baseline in Table 2.

### D.   The ICE method, in effect, re-mixes phrases from previous observed attacks.

The reconstruction task used to train the generator in the ICE method (Alg. 1) often leads to exact memorization of train-set examples or exact phrases from the train-set. Qualitatively, we find that increasing the beam search parameter $\alpha$ and number of steps aligning the steering signal, or decreasing reconstruction loss, $\lambda$ at inference time leads to remixing these phrases (See Table 1). This re-mixing effect also explains the high MAUVE distributional similarity between the ICE methods and the ANLI rounds on which it trains.

Chan et al. (2022) suggests that LLMs learn concepts better when they are repeated, esp. when they are repeated in slightly varied contexts; ICE's re-mixing behavior facilitates exactly this outcome. We hypothesize that the repetition and slightly varied contexts may help the model learn the essence of the original human attacks, leading to improved performance against similar attacks.

## 7   Limitations

If we put our new more robust models to use, human adversaries may adapt to them as well. Checking whether crowd-sourcing fresh attacks is indeed more difficult on the new models is beyond the scope of this work. Also, we benefit from having a fair number of human adversarial examples (16.9$k$ in ANLI, and 10$k$ per round in Hate Speech). Our methods are less successful in a scenario with much fewer examples ($\sim 100$). On the flip side, we have also not evaluated these methods on classifiers with access to even larger real adversarial datasets.

We ran our experiments on popular transformer architectures: BERT-Large for classification (as this is the model that was attacked to create the adversarial datasets we rely upon) and T5 for generation. T5 was chosen for two reasons: 1) it is comparable to BERT-Large in size and performance but uses a different architecture and training, thus ensuring our findings are not just due to model size difference. 2) The transformer Encoder-Decoder is now the ubiquitous generative model architecture. The generalization of our findings to non-transformer architectures is left for future work.

Finally, our methods work on datasets with the notion of an original example, and a perturbed adversarial example as is the norm for adversarial robustness literature (Madry et al., 2017). In the new paradigm of larger more capable NLP models, adversarial datasets may increasingly not involve a perturbation (Ganguli et al., 2022).

**Risks:** Our techniques can enhance robustness given a set of observed adversarial examples. The new classifier we trained with generated data from DI and ICE may still be vulnerable to future human attacks that are able to adapt to the new model (in this paper future attack rounds are known a priori from past model-in-the-loop work). This would require extensive crowd-sourcing efforts to evaluate. We also run the risk of over-fitting to the new human generated adversarial data. This may come at the cost of lower performance on future attacks generated by a different mechanism (say TextFooler instead of future ANLI rounds), and comes at the cost of degrading accuracy on original tasks such as MNLI and SNLI (Table 19 in Appendix). As a separate concern, any technique that betters generative text modeling brings the risk that humans may

struggle to distinguish machine generated text. This can have negative consequences for disinformation and misinformation, which is an active area of research (Pu et al., 2023).

## 8 Conclusion

We demonstrate that training on attacks that imitate human adversaries can improve robustness to future rounds of human adversarial attacks by 11% on ANLI, and by 6% and 8% on existing adversarial examples on ANLI and hate speech datasets. We are able to improve robustness to future attack distributions even when the attack generator is only trained on 1000 real adversarial examples. We show that existing attack generation methods that do not train on the distribution of real attacks, however, (methods like TextFooler and CT-GANs) are unable to improve robustness to future real attacks. Finally, we discover that attack generation methods with the lowest label noise or highest attack success rate or highest distributional similarity to future attacks are not the best methods at increasing robustness to future real attacks. These findings run counter to accepted norms on choosing the best attack method type in literature, demonstrate the opportunities in leveraging increasingly effective generation methods, and motivate future work on improving real adversarial robustness.

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

# A   Appendix

## TextFooler: Generated Examples

A random selection these attacks on round 1 of the ANLI dataset are included in Table 9. A random selection of these attacks on the first round 2 (Analogous to round 1 in ANLI) of the Hate Speech Detection dataset are included in Table 13. Trends for this attack type are explained in the table captions.

## BertAttack: Generated Examples

A random selection these attacks on round 1 of the ANLI dataset are included in Table 7. A random selection of these attacks on the first round 2 (Analogous to round 1 in ANLI) of the Hate Speech Detection dataset are included in Table 14. Trends for this attack type are explained in the table captions.

## CT-GAN: Generated Examples

A random selection these attacks on round 1 of the ANLI dataset are included in Table 8. Trends for this attack type are explained in the table captions.

## DI method: Generated Examples

A random selection these attacks on round 1 of the ANLI dataset are included in Table 10. A random selection of these attacks on the first round 2 (Analogous to round 1 in ANLI) of the Hate Speech Detection dataset are included in Table 15. Trends for this attack type are explained in the table captions.

## ICE method: Generated Examples

A random selection these attacks on round 1 of the ANLI dataset are included in Table 11. A random selection of these attacks on the first round 2 (Analogous to round 1 in ANLI) of the Hate Speech Detection dataset are included in Table 12. Trends for this attack type are explained in the table captions.

## Implementation Details for DI Method.

In this section we give more specifics needed to replicate the DI method. The general outline of the method is provided in Section 4.

For the generator, we use the default settings for the base T5 model, with no changes to the hyper-parameters (Raffel et al., 2020)).The base T5 model is then trained on ANLI R1 attacks, with the premise and the label as the input (delimited by a special token [NLI_LABEL]:). The label is denoted by one of the three letters: (e, n, c) to denote entailment, neutral, and contradiction. The output sequence that the T5 model is trained to decode is the attack hypothesis in the ANLI R1 data. We use the standard cross-entropy loss function as described in the base T5 model and use the train split of the ANLI R1 data. We choose the best checkpoint based on the token-level accuracy in generating the hypotheses in the validation set of the ANLI R1 dataset. We then produce an in-sample generated attack dataset by producing multiple hypotheses (up to 100) on the train split of ANLI R1. Thus, we ensure that there is no leakage of attacks that were not already previously seen by the base classifier.

The cross-entropy loss of the generated text probabilities $\widehat{\mathbf{X}_{\mathbf{a}_0}}(x_i, y)$ and the target text $x_o$, assuming a maximum sequence length $K$, and vocabulary $\mathbf{V}$ is given by:

$$\mathcal{L}_{ce}(x, y) = -\sum_{k=1}^{K} \sum_{v \in \mathbf{V}} \mathbb{1}_{x_{o,k}=v} \log(\widehat{\mathbf{X}_{\mathbf{a}_0}}(x_i, y)_{k,v}) \tag{3}$$

For the classifier, we use the BERT-Large pre-trained model, and first fine-tune it on the MNLI+SNLI data mixture as per the ANLI benchmark. We then fine-tune that classifier further on the ANLI R1 train

| Original ANLI R1 Premise | Original ANLI R1 Hypothesis ($\mathbf{a_0}$) | BertAttack(R1): Generated Hypothesis ($\mathbf{a_g}$) | Original Label | Label Still Correct? | Base ($\widehat{y}_{\theta_0}$) | Base + R1 ($\widehat{y}_{\theta_1}$) | Base + R1 + R2 ($\widehat{y}_{\theta_2}$) |
|---|---|---|---|---|---|---|---|
| The Other One is the third solo album by former Fleetwood Mac guitarist Bob Welch. The track "Future Games" was first released on the Fleetwood Mac album of the same name in 1971. Members of Welch's backing band also make songwriting contributions here though the majority of tracks are Welch's own | Members of Welch's backing band include Maynard James Keenan who assisted Bob Welch with almost every track on The Other One." | Member of Welch's backing band encompass Menard James Keenan who helped Boba Welsh with about everything trails on The Else One | e | n | n/a | n/a | n/a |
| The Walloon Legion (French: "Légion Wallonie" ) was a collaborationist volunteer unit recruited from Belgium's French-speaking population in Wallonia and Brussels during the German occupation of World War II. The Walloon Legion served in the Wehrmacht, later in the Waffen-SS, on the Eastern Front on both front line and reserve duties. | The Walloon Legion was partly recruited from Brussels. | The Walloon Legion was not a volunteers units. | e | n | n/a | n/a | n/a |
| Irma Pezzia Haubold (November 20, 1908 – April 4, 1996) was an American artistic gymnast. She competed at the 1936 Summer Olympics and placed fifth with the team. She was married to a fellow Olympic gymnast Frank Haubold. They were the first married couple of compete in the same Olympics. | Irma competed in the 1940 Summer Olympics. | Irma participated in the 1940 Hsia Olympics. | c | y | e | c | c |
| Mia Foni is the debut album of Greek American singer Annet Artani. It features 19 tracks in both Greek and English, including "Why Angels Cry", the song that Annet performed in the Eurovision Song Contest 2006 in Athens representing Cyprus. The album was released in both Greece and Cyprus where it entered the top 10. | Annet Artani's album have 15 tracks. | Annet Artani's albums have 15 trajectories., | c | y | e | e | c |
| Ernest Thompson Willows (1886–1926) was a pioneer Welsh aviator and airship builder. He became the first person in the United Kingdom to hold a pilots certificate for an airship when the Royal Aero Club awarded him "Airship Pilots Certificate No. 1 | Ernest Thompson Willows was a pilot. | Ernesto Thomson Willows was a experimenter | n | y | c | c | n |
| James Hugh Sinclair (16 October 1876 – 23 February 1913) was a South African cricketer who played in 25 Tests from 1896 to 1911. He scored South Africa's first three Test centuries and was the first person from any country to score a century and take five wickets in an innings in the same Test. He is one of the fastest-scoring Test batsmen of all. | They refered to him as a fast scorer. | They refered to him as a rapids scorer., | e | y | n | e | e |

Table 7: Selected examples from BertAttack(R1) attacks. Like the TextFooler attacks in Table 9 this attack makes small modifications to the hypothesis, explaining the high level of distributional similarity to ANLI rounds 1 and 2. Unlike TextFooler, however, BertAttack makes more sophisticated contextual phrase substitutions. For example in row 2, "partly recruited from Brussels" is replaced with "not a volunteers units."

data split. We choose the checkpoint based on the best validation set accuracy of ANLI R1 dataset. This fine-tuned model forms the basis of our comparison, and our goal is to improve adversarial robustness beyond what this model can achieve.

In the DI method, we then fine-tune the Base + R1 model on the generated examples. For all fine-tuning processes, we used an adamw optimizer in Jax + Tensorflow. The following learning rate schedule was used: There are first 3,681 warm-up steps at the initial learning rate of 3.0e-05. Then for the next 36,813 steps the learning rate decays at a linear rate (Though we only fine-tune for 40k steps total). The checkpoint that performs best on the validation split is selected for the next stage.

| Original ANLI R1 Premise | Related Original ANLI R1 Hypotheses ($\mathbf{a_0}$) | CT-GAN(R1): Generated Hypothesis ($\mathbf{a_g}$) | Original Label | Label Still Correct? | Base ($\widehat{y}_{\theta_0}$) | Base + R1 ($\widehat{y}_{\theta_1}$) | Base + R1 + R2 ($\widehat{y}_{\theta_2}$) |
|---|---|---|---|---|---|---|---|
| The Parma trolleybus system (Italian: "Rete filoviaria di Parma" ) forms part of the public transport network of the city and "comune" of Parma, in the region of Emilia-Romagna, northern Italy. In operation since 1953, the system presently comprises four urban routes. | "Since 1953 the Parma Troleybus system has been comprised of four urban routes." | Parma has been in operation for the entire region of Emilia-Romagna, northern Italy. | e | y | e | e | e |
| Spoons was a comedy sketch show first broadcast on the United Kingdom's Channel 4 from 30 September 2005. In the United States, "Spoons" is broadcast on BBC America. The relationship themed show combined recent trends in sketch shows—dark content, strong language, and recurring catchphrases. | Spoons the comedy sketch show first appeared on the United Kingdom's Channel 4 a year after 2005. | Spoons was a comedy sketch show first broadcast on the United Kingdom's Channel 4 from 30 September 2005 | c | n | n/a | n/a | n/a |
| The 1992 Boise State Broncos football team represented Boise State University in the 1992 NCAA Division I-AA football season. The Broncos competed in the Big Sky Conference and played their home games at Bronco Stadium in Boise, Idaho. Led by sixth-year head coach Skip Hall, Boise State finished the season 5–6 overall and 3–4 in conference. | ["After 1991 Boise State Broncos competed in the Big Sky Conference.", "Boise State won four games at their home stadium in 1993.]" | The 1992 Boise State Broncos football team competed in the Big Sky Conference and played their home games at Bronco Stadium. | e | y | e | e | e |
| Other ranks (or ORs) in the Royal Danish Army, Royal Danish Navy and Royal Danish Air Force is a joined term for military personnel that are not officers of various forces, by the NATO system of ranks and insignia. These personnel are NCOs and privates, seamen or aircraftsmen | There are no NCOs, non-commissioned officers, in the Danish Military services. | These personnel are NCOs and privates, seamen or aircraftsmen. | c | n | n/a | n/a | n/a |
| "Drunk Girls" is a song by American rock band LCD Soundsystem. It was released as the second single from their third studio album, "This Is Happening" (2010), on May 3, 2010. Band frontman James Murphy has described the song as "dumb" but added "I like dumb, short stuff." The 7" single features a cover of the song by San Francisco psychedelic rock band Wooden Shjips. | James Murphy has said "I like dumb, short stuff". | "Band frontman James Murphy has said "I like dumb, short stuff". | e | y | e | e | e |
| Viru is a 5.0% ABV pilsner-style beer brewed in Estonia. It is brewed in the country's second largest city, Tartu, by the A. Le Coq brewery. The brand is owned by Baltic Beer Company Ltd (formerly Brand Independence Ltd), based in London, UK, and is brewed under licence in Estonia. A. Le Coq is the second largest brewery in Estonia, with a market share of 36.8% in 200 | Viru is a 5.0% ABV pilsner-style beer brewed in Estonia. It is brewed in the country's second largest city, Tartu, by the A. Le Coq brewery. Lately there have been talks of setting up another brewery in the same region | Viru is brewed in the country's second largest city, Tartu, by the A. Le Coq brewery | n | n | n/a | n/a | n/a |

Table 8: Selected examples from CT-GAN(R1) attacks. These attacks most often are direct sub-phrases of the premise. While they generate the entailment class examples correctly, almost no new information is ever added. Meanwhile, the tendency to pick up sub-phrases from the premise means the other two classes, neutral and contradiction, are largely generated incorrectly (See row 2 and 4). Further, since these incorrect example are often a verbatim copy of the premise, we may be teaching the model to not mark entailment for content directly in the premise).

**Implementation Details for ICE Method.**

In this section we give more specifics needed to replicate the ICE method. The general outline of the method is provided in Section 4, and Alg. 2 provides more detail.

---
**Algorithm 2** Pseudo-code for ICE Method (in more detail)

---
**Part 0: Define Variables.**

We break the input $x$ into two halves, $x = (x_i, x_o)$. Let $x_o$ be the portion of the text example that is perturbed by the attacker, and $x_i$ be the unchanging context. Let $y$ be the true label corresponding to $(x_i, x_o)$ for a given NLP task.

Let `ReconstructToken` and `TaskToken` be fixed constant tokens that identify which task we are asking $T$ to perform.

**Part 1: Set up components.**

Let $T$ be an encoder decoder transformer model.

Let $S$ be a vector of the activation states of the hidden layer after each transformer unit in $T$. This is the steering signal. We use $S(i, o)$ to denote the signal when we pass input (`ReconstructToken`, $i$) into $T$ and $T$ has generated $o$.

Let $C$ be a simple linear feed-forward classifier that takes $S$ as input.

**Part 2: Warmstarting Trick.**

Fine-tune $T$ on dataset $\mathbf{a}_0$ on the reconstruction task: (`ReconstructToken`, $x_i$) $\rightarrow x_o$, $\forall (x, y) \in \mathbf{a}_0$.

Fine-tune $T$ on dataset $\mathbf{a}_0$ to predict the true label for the NLP task: (`TaskToken`, $x$) $\rightarrow y$, $\forall (x, y) \in \mathbf{a}_0$.

Freeze the parameters of $T$ so that they are not updated.

Train $C$ on on the dataset $\mathbf{a}_0$ to predict the true label for the NLP task: $S(x, \emptyset) \rightarrow y$, $\forall (x, y) \in \mathbf{a}_0$.

**Part 3: Generate synthetic examples.**

Let $T'$ be a saved copy of $T$.

Unfreeze parameters of $T$ so that they may be updated.

Freeze the parameters of $C$ so they do not change.

Since $S$ is a function of the hidden layers of $T$, it may also change as the parameters of $T$ change.

**for all** $(x, y) \in \mathbf{a}_0$ **do**

    Pass in (`ReconstructToken`, $x_i$) into $T$ as input.

    Let $a_g$ be the output generated by $T$. Set $a_g = \emptyset$.

    **while** $T$ has not generated end of sentence token or hit maximum length for $a_g$ **do**

        **for all** $k \in 1, 2, \cdots, 10$ **do**

            `ReconstructionLoss` = Cross entropy loss between $a_g$ and $x_o$.

            `ClassifierTaskLoss` = Cross entropy loss between $C(S(x, a_g))$ and $y$.

            `Loss` = `ClassifierTaskLoss` $+ \lambda$ `ReconstructionLoss`

            Update parameters of $T$ based on the $\nabla$`Loss`.

        **end for**

        Use $T$ to generate the next token, and append the new token to $a_g$.

    **end while**

    **yield** $a_g$, a new synthetic example.

    Set $T = T'$

**end for**

---

| Original ANLI R1 Premise | Original ANLI R1 Hypothesis ($\mathbf{a}_0$) | TextFooler(R1): Generated Hypothesis ($\mathbf{a_g}$) | Original Label | Label Still Correct? | Base ($\widehat{y}_{\theta_0}$) | Base + R1 ($\widehat{y}_{\theta_1}$) | Base + R1 + R2 ($\widehat{y}_{\theta_2}$) |
|---|---|---|---|---|---|---|---|
| "Aama" (Literally: Mother) is the first Nepali movie produced in Nepal, starring Shiva Shankar Manandhar and Bhuvan Chand (Thapa) as the leading actors. The movie was produced by the Information Department of the Nepalese Government and released on October 7, 1964. Bollywood film maker Hira Singh Khatri was invited by the late King Mahendra to direct the first Nepali movie. | Hira Singh Khatri was the director for the movie "Aama". | Hira Seng Khatri was the headmaster for the film "Aama". | "e" | False | n/a | n/a | n/a |
| "Dangerous" is a song by American electronic music project Big Data, from their debut EP "1.0" (2013) and their debut studio album "2.0" (2015). It features American indie rock band Joywave, with vocals being performed by the band's lead singer Daniel Armbruster. | Big Data did not produce any albums in 2007. | Massive Data did not engender any albums in 2007. | "e" | False | n/a | n/a | n/a |
| Camp Al-Saqr, referred to by some media sources as Camp Falcon, Forward Operating Base Falcon, Joint Service Station (JSS) Falcon, or Combat Outpost Falcon, was a United States military forward operating base in Iraq a short distance outside Baghdad, some 13 km south of the Green Zone. In OIF 2004; it was designated as "Camp Ferrin-Huggins". s of 2009 , the base housed up to 5,000 troops. | None of the soldiers agreed with the names given to the camp. | None of the soldiers countersigned with the names gave to the campground. | "n" | True | "e" | "e" | "e" |
| Wolf hunting with dogs is a method of wolf hunting which relies on the use of hunting dogs. While any dog, especially a hound used for hunting wolves may be loosely termed a "wolfhound", several dog breeds have been specifically bred for the purpose, some of which, such as the Irish Wolfhound, have the word in their breed name. | wolves are hunting dogs | wolfe are hunting dogs | "c" | True | "e" | "n" | "e" |
| The discography of American metalcore band As I Lay Dying consists of 6 studio albums, 2 compilation albums, 1 video album, 11 singles and 15 corresponding music videos as well as 1 split album with fellow metalcore band American Tragedy called "As I Lay Dying/American Tragedy". | The bands As I Lay Dying has filmed over 5 dozen music videos. | The strips As I Lay Death has videotaped over 5 dozen music videos. | "c" | False | n/a | n/a | n/a |
| Sir John Peebles Arbuthnott, PPRSE, FRCPSG, FMedSci, FRCPath (born 8 April 1939) is a Scottish microbiologist, and was Principal of the University of Strathclyde. He succeeded Lord Wilson of Tillyorn as President of The Royal Society of Edinburgh in October 2011 and was succeeded by Dame Jocelyn Bell Burnell in October 2014. | Sir John Peebles started out as a teacher. | Monsieur John Peebles commenced out as a teacher. | "n" | True | "e" | "e" | "e" |

Table 9: Selected examples from TextFooler(R1) attacks. These attacks make small random modifications to the hypothesis, explaining the high level of distributional similarity to ANLI rounds R1 and R2. More examples are included in Supplemental Materials.

In addition to the DI method setup, we train a linear classifier using the T5 generator. The linear classifier takes in each hidden layer at the end of each transformer unit. We flatten and concatenate the hidden layers before feeding them into a single dense layer + layer norm with 3 as the output dimension size (one for each class in NLI). This is a departure from the PPLM implementation which sums the hidden layers (Dathathri et al., 2020). Adding another layer, even a small 3 by 3 dense layer, does help improve this linear classifier but can slow down the attack generation process. So, we did not use any additional layers. The linear

classifier still does not perform as well as T5 or BERT-Large on the NLI task (71% accuracy on MNLI dev as compared to 91% for T5 and 89% for BERT-Large).

Assuming that the set of labels $\mathbf{m}$ : {entailment, neutral, contradiction}, the classifier optimizes the following cross-entropy classification loss:

$$\mathcal{L}_c(x, y) = -\sum_{m \in \mathbf{m}} y \cdot \log(\texttt{LC}(H(x))_m) \tag{4}$$

Further, the reconstruction loss used to warm-start the generator is given by:

$$\mathcal{L}_r(x, y) = -\sum_{k=1}^{K} \sum_{v \in \mathbf{V}} \mathbb{1}_{x_{o,k}=v} \log(\widehat{\mathbf{X}_{\mathbf{a}_0}}(x, y)_{k,v}) \tag{5}$$

During attack generation, we use a high $\alpha$ beam parameter, 0.7 for ANLI, and 0.8 for the Toxicity dataset. For additional example diversity, the weight on the reconstruction loss can be made negative: results here are presented with a reconstruction loss weight of 0 for ANLI, and $-1.5$ for the Toxicity dataset. On each step that we update the T5 parameters, we smooth the update (final parameter = original parameter * 0.75 + new parameter * 0.25). We did not substantially explore these hyper-parameters, as each example generation run takes $\sim 100$ GPU hours from warm-up steps to example generation.

We only include generated attacks if the linear classifier labels the generated example belongs to the original class. For example, we filter attacks that contain words in the set { "e", "n", "c" }, or more than two of each of those class labels (filtering "eee", or "cccc"), as these are NLI labels and an artifact of the NLI pretraining. We also filter attacks that exactly match a hypothesis in the observed set of examples, attacks that are less than two words long, or attacks that repeat a string of four characters more than four times (to avoid artifacts like: "cococococococococococococo" that may appear).

If the ICE method is used to generate data from a new round, (ex: R1 for ANLI), the base model is trained on the new round and synthetic data all in the same fine-tuning period (randomly shuffling the data). The original data is repeated until it is equal in number to the synthetic data. For example, if there are $1k$ original examples, and $5k$ generated data, the original set of $1k$ will be repeated five times, and the final shuffled dataset will have $10k$ examples. This ratio and shuffling method could be tuned further, but we did not attempt this.

### Implementation Details for CT-GAN.

To implement the CT Generative Adversarial Network, adapted to the ANLI task, we use the TF-GAN library [1] with a generator and discriminator initialized with the T5 and the BERT-Large model fine-tune on MNLI, SNLI, ANLI R1 data in a sequential manner. These initializations are inline with what we use in our DI and ICE methods. Then we train the generator and discriminator in an iterative manner. In the first step, we fine-tune the generator to generate examples similar to ANLI R1 and produce a misclassification error from the classifier using a modified min-max objective. In the next step, we further fine-tune the classifier on those generated examples, with the original label as that of the corresponding ANLI R1 examples. We then repeat these two steps 10 times. In each step, we train the generator for 20,000 steps, and the discriminator for 10,000 steps.

### Implementation Details for TextFooler.

To generate text attacks based on word-substitution, we use the TextFooler recipe outlined in (Jin et al., 2020). We use the implementation in the TextAttack library [2] and produce augmentation examples by perturbs the ANLI R1 hypothesis through word-embedding swap. We generated 50 transformations per example, inline with the amount of data generated using our imitation and exploration based methods.

---

[1] https://blog.tensorflow.org/2019/08/introducing-tf-gan-lightweight-gan.html
[2] https://github.com/QData/TextAttack

**Implementation Details for fine-tuning BERT Large.**

We used the default parameters for the uncased BERT Large model detailed in the corresponding paper, (Devlin et al., 2019). The checkpoint that performed best on the ANLI dev set was used.

**Human Rating guidelines.**

The paper authors are the human raters. These are the guidelines we follow for edge cases.

- The current year is assumed to be 2019, since we are working with the base T5 model trained in that year, and our dataset, ANLI is also generated in that year.

- Examples with incorrect grammar are not penalized if the matter asserted fit the correct label. (ex: "They is a very popular band")

- Mild spelling mistakes, where the meaning is absolutely clear after reading the premise are not penalized: (example: FeliCa is a intelligent card system developed for utilizing in *electron* cash cards but has not yet gain endorsement to be utilizing in the United States).

- Spelling mistakes in named entities create a new entity (ex: "Kënga" is not the same as "K??nga" and "Mars" is not the same as "Mar").

- Word substitutions from other languages are okay if they are commonplace English knowledge (ex: "sir" –> "monsieur"), but not okay if they change the meaning of the sentence (ex: "life" –> "vie", which has an alternative English meaning).

- Repeating phrases is okay. (ex: Mark Donovan best known for his role in productions such as "Shaun of the Dead", "Shaun of the Dead", "Black Books", "In Bruges", and "Murder Investigation Team.") since they don't change the truth of the matter asserted.

- Tautologies are still considered "entailment" if they are true given the premise. (Ex: "The movie, The Golden Compass, is a movie.")

**Additional ANLI Result: Holding out R3 as a future round and training on R1 and R2 yields similar conclusions**

Table 17 extends the setting from Table 2 where R1 and R2 are past observed attacks available for training, and R3 is the held out set of future human attacks. All conclusions are the same as the ones drawn from Table 2.

**Additional Result: Size of $a_g$ used to train $\widehat{y}_{\theta_1}$: More examples are better up until a point.**

Here, we seek to understand the role of the number of generative adversarial examples used for fine-tuning the base classifier. We use all available ANLI R1 real adversarial examples for fine-tuning the adversarial generator, and decode the specified number of examples from the generator for fine-tuning the base classifier. We demonstrate that as the number of generated examples increase, adversarial robustness increases up to a point (up to 50x: 50 generated examples for every real adversarial example). After that point, we observe more generated examples impacts adversarial robustness negatively as shown in Table 16.

For the 50x experiment, however, we observe the best checkpoint while training BERT-Large is usually reached when the model has trained on only $\sim 30\%$ of the generated examples provided. Therefore these findings are inconclusive. This behavior could be due to the interaction between (1) increased label noise/diversity of examples generated as we increase the number of generated examples due to a quirk of the decoding process, and/or (2) the improvement in adversarial robustness due to more examples incorporated into training until overfitting to the generated distribution.

**Sanity Check: More fine-tuning on real adversaries does not help.**

We study whether the gains obtained by the generated examples could have been obtained otherwise merely by training longer on the ANLI R1 data itself. In Table 18, we see that if we had continued for more steps training on R1 data, we do not observe higher validation set accuracy as compared to training on training on DI(R1) generated attacks. (These figures are not directly comparable to 2 in the Results section of the main paper, since here we present validation set accuracy, and the Results section contains test set accuracy).

**Correlation analysis of heuristics with robustness**

In Table 21, we find that the Pearson correlation observed between accuracy achieved on the three rounds of ANLI and the corresponding dataset heuristics those models were trained on is not significant. These correlation coefficients were computed to understand if any adversarial dataset's heuristic – attack success, label noise or distributional similarity correlates with how well a classifier would perform on future rounds of human adversarial attacks. Our findings suggest that none of these heuristics are indicative and future work could focus on finding such a data quality measure that correlates with future human adversarial robustness.

| Original ANLI R1 Premise | Original ANLI R1 Hypothesis ($\mathbf{a_0}$) | DI(R1): Generated Hypothesis ($\mathbf{a_g}$) | Original Label | Label Still Correct? | Base ($\widehat{y}_{\theta_0}$) | Base + R1 ($\widehat{y}_{\theta_1}$) | Base + R1 + R2 ($\widehat{y}_{\theta_2}$) |
|---|---|---|---|---|---|---|---|
| The Coward is a 1915 American silent historical war drama film directed by Reginald Barker and produced by Thomas H. Ince. Ince also wrote the film's story and scenario with C. Gardner Sullivan. The film stars Frank Keenan and Charles Ray. John Gilbert also appears in an uncredited bit part. A copy of "The Coward" is preserved at the Museum of Modern Art. | The Coward is a part of American history and located in a museum for preservation. | The Coward is a 1915 American silent historical war drama film The Coward is a 1915 American silent historical war drama film | "e" | True | "c" | "c" | "c" |
| "Dangerous" is a song by American electronic music project Big Data, from their debut EP "1.0" (2013) and their debut studio album "2.0" (2015). It features American indie rock band Joywave, with vocals being performed by the band's lead singer Daniel Armbruster. | Big Data did not produce any albums in 2007. | It is unlike any other Electronic Music release, but the Joywave may have been proud of us with our music. | "e" | False | n/a | n/a | n/a |
| The Joint Aviation Authorities (JAA) was an associated body of the ECAC representing the civil aviation regulatory authorities of a number of European States who had agreed to co-operate in developing and implementing common safety regulatory standards and procedures. It was not a regulatory body, regulation being achieved through the member authorities. | The JAA is not a regulatory body. | authorities of a number of European states are from civil aviation authorities and represent the civil aviation authorities of a number of European States who had agreed to regulate these standards and safety regulations. | "e" | False | n/a | n/a | n/a |
| 11 Flowers () is a 2011 film directed by Wang Xiaoshuai centered on a boy named Wang Han who loses his white shirt which shows that he is his school's best gymnast. The loss of the shirt is to take on greater metaphorical meaning as the film progresses which is inspired by the director's own experience as a youth during the cultural revolution and the more general confusion of childhood. | Wang Xiaoshuai directed 11 films in 2011. | The loss of the shirt is to take on great film productions as the film progresses when it progresses to the end of the film. | "n" | False | n/a | n/a | n/a |
| Tiffanie DeBartolo (born November 27, 1970) is an American novelist, filmmaker, and co-founder of independent record label Bright Antenna. She has written two novels, "God-Shaped Hole" and "How To Kill a Rock Star". She wrote and directed the film "Dream for an Insomniac", featuring Jennifer Aniston, but which had a very limited release in 1996. | Jennifer Aniston was not an actress in any movie directed by Tiffanie DeBartolo. | The film "Dream for an Insomniac" written and directed by Tiffanie DeBartolo was widely released in 1994. | "c" | True | "n" | "n" | "n" |
| Viru is a 5.0% ABV pilsner-style beer brewed in Estonia. It is brewed in the country's second largest city, Tartu, by the A. Le Coq brewery. The brand is owned by Baltic Beer Company Ltd (formerly Brand Independence Ltd), based in London, UK, and is brewed under licence in Estonia. A. Le Coq is the second largest brewery in Estonia, with a market share of 36.8 | A. Le Coq is the second largest brewery in Estonia, with a market share of 36.8% in 2005. It had an increased market share in 2015. | A. Le Coq is the second largest brewery in Estonia, with a market share of 36.8% in 2005. The A. Le Coq brewery makes a lot of beer in Latvia, which had a market share of 36.8% in 2005. | "n" | True | "e" | "e" | "e" |

Table 10: Selected examples from DI(R1) attacks. These generated hypotheses deviate pretty significantly from the original hypothesis. They also incorporate knowledge from the premise (ex: "second largest" phrase is common to the premise and generated hypothesis in the Le Coq example on the bottom row). More examples are included in Supplemental Materials. Unlike the other methods, even when the label is incorrect, the generated hypothesis is usually still a pretty cogent sentence.

| Original ANLI R1 Premise | Selected Original ANLI R1 Hypotheses for this premise ($\mathbf{a_0}$) | ICE(R1): Generated Hypothesis ($\mathbf{a_g}$) | Desired Label | Label Correct? | Base ($\widehat{y}_{\theta_0}$) | Base + R1 ($\widehat{y}_{\theta_1}$) | Base + R1 + R2 ($\widehat{y}_{\theta_2}$) |
|---|---|---|---|---|---|---|---|
| The Coward is a 1915 American silent historical war drama film directed by Reginald Barker and produced by Thomas H. Ince. Ince also wrote the film's story and scenario with C. Gardner Sullivan. The film stars Frank Keenan and Charles Ray. John Gilbert also appears in an uncredited bit part. A copy of "The Coward" is preserved at the Museum of Modern Art. | ["Reginald Barker directed a movie in 1915.", "A 1915 American silent historical war drama film is preserved at the Museum of Modern Art.", ···] | Reginald Barker movie was a silent drama. | "e" | True | "c" | "c" | "c" |
| 11 Flowers () is a 2011 film directed by Wang Xiaoshuai centered on a boy named Wang Han who loses his white shirt which shows that he is his school's best gymnast. The loss of the shirt is to take on greater metaphorical meaning as the film progresses which is inspired by the director's own experience as a youth during the cultural revolution and the more general confusion of childhood. | ["The movie was inspired by the director's experience as a youth during the american revolution.", "11 Flowers () is a 2011 film directed by Wang Xiaoshuai about boy named Wang Han who loses his white shirt and is no longer a top gymnast. The shirt becomes a metaphor for childhood and growing up in China.", ···] | 11 Flowers () is a 2011 film directed by Wang Xiaoshuai about boy named Wang Han who loses his white shirt and is top gymnast. The shirt becomes a metaphor for the director's experience of childhood. The shirt becomes a metaphor for the directors's experience of childhood.. The shirt becomes a metaphor for the film's story. | "e" | True | "c" | "c" | "c" |
| Domestically, he has played for Khan Research Laboratories, Lahore, Lahore Eagles, Lahore Lions, Lahore Qalandars, Pakistan A and Huntly (Scotland) during his career. He was the captain of Lahore Qalandars in the first edition of the Pakistan Super League. | ["There are multiple teams based in Lahore, Pakistan.", "There has been at least one domestic sports league in Pakistan.", ···] | There are at least several teams in life | "n" | True | "n" | "n" | "n" |
| Benjamin A. Muncil (28 Aug 1867 – 16 Dec 1930) was an American master builder in the Adirondacks early in the 20th century. He was a major figure in the architectural development of the Adirondack Great Camps; among his many projects was Marjorie Merriweather Post's Camp Topridge, Northbrook Lodge, and White Pine Camp, a summer White House of US President Calvin Coolidge. | ["Benjamin Muncil has built many things", "One of Muncil's projects was created for a former US President. ", ···] | One of the best American master builder in the Adirondacks projects built many things created for a former US President. | "e" | True | "e" | "e" | "e" |
| Piton is a Pilsner beer brand from the island of Saint Lucia, brewed by Windward & Leeward Brewing Limited, which is owned by Heineken. The beer was named for the Gros Piton and Petit Piton mountains on the island. It was first brewed on October 7, 1992. | ["Even though the beer was first brewed in public in October, many tasted it previously", "Even though the beer was brewed in October, it is not certain where that occurred ", ···] | Even though the beer was brewed in public in October, no one tasted it previously the first of taste it formerly | "n" | False | n/a | n/a | n/a |
| Benjamin A. Muncil (28 Aug 1867 – 16 Dec 1930) was an American master builder in the Adirondacks early in the 20th century. He was a major figure in the architectural development of the Adirondack Great Camps; among his many projects was Marjorie Merriweather Post's Camp Topridge, Northbrook Lodge, and White Pine Camp, a summer White House of US President Calvin Coolidge. | ["Benjamin Muncil has built many things", "One of Muncil's projects was created for a former US President. ", ···] | One of the best American master builder in the Adirondacks projects built many things created for a former US President. | "e" | True | "e" | "e" | "e" |

Table 11: Selected examples from ICE(R1) attacks. This method generates hypotheses that learn from past seen hypotheses and the premise. Common failure modes include generating nonsensical sentences or getting stuck in loops of the same phrase (see: row 2). Some of the neutral labeled generations are technically correct, but very random (see: row 3). More examples with a complete set of original hypotheses are included in Supplemental Materials.

| Original Hate Speech R2a Example($\mathbf{a_0}$) | Matched Hate Speech R2b Example($\mathbf{a_0}$) | ICE(R2): Generated Example ($\mathbf{a_g}$) | Desired Label | Label Correct? | Base(R1) ($\widehat{y_{\theta_0}}$) | Base(R1) + R2 ($\widehat{y_{\theta_1}}$) |
|---|---|---|---|---|---|---|
| are black folks extinct? cause they should be (hate) | are black panthers extinct? (not hate) | man black folks always wear black. | "h" | False | "h" | "h" |
| lesbian r subhummmmmman (hate) | Why the fuck do you think lesbians are subhuman? (not hate) | lesban facias doesn't deserve rights. | "h" | True | "h" | "h" |
| Who would cut off their wiener for a hole...dont get it. it's like my brain just farts when I try to think of what must go through these people's heads to ever contemplate making that sort of crazy choice (hate) | Who would cut their wiener in several pieces for lunch..dont get it. it's like my brain just farts when I try to think of what must go through these people's heads to ever contemplate making that sort of crazy choice. (not hate) | Who would cut off their armies? | "n" | True | "h" | "n" |
| I can believe that she thinks arabs are the most kind ethnic group, like come on they are lovely (not hate) | I cant believe that she think arabs are the most kind ethnic group, like come on more like least (hate) | how can he believe he cannot teach anyone so much English | "n" | True | "n" | "n" |
| black people are super clean (not hate) | negroes are not clean (hate) | Black people are BLIGGNOW | "n" | True | "h" | "h" |
| My work doesn't cater for people with invisible disabilities. (not hate) | Glad my work doesn't cater for people with disabilities. (hate) | My work doesn't cater for pet dogss who're close to pet parents. | "n" | True | "h" | "h" |
| look dude, gyps are evil and basically cant be trusted. Especially with their cheating crystal balls, they freak me the fuck out. Grubby looking caravan lovers. (hate) | look dude, they are evil and basically cant be trusted. Especially with their cheating crystal balls, they freak me the fuck out. Grubby looking witches. (not hate) | look dude, chinese climates are evil and basically cant be trusted Especially with chinese chia | "h" | False | "h" | "h" |
| your personality is so ugly it could use makeup, typical black chick (hate) | your personality is so ugly it could use makeup (not hate) | Your black for black for a black person is too heavy for whites | "h" | True | "h" | "h" |
| all women are not to be trusted, they are snakes, vicious vipers trying to trick their men (hate) | all men are not to be trusted, they are snakes, vicious vipers trying to trick their women (not hate*) | all women are bitches everyone they have all fucking wifes | "h" | True | "h" | "h" |
| the bathroom is scummy we need some viakal and to get the cleaning sqaud in here. Maybe I'll do that tomorrow, ima go sort out somet stuff now (not hate) | the bathroom is so scummy it looks like an Arab house, honestly. We need some viakal and to get the cleaning sqaud in here. Maybe I'll do that tomorrow, ima go sort out somet stuff now (hate) | the bathroom is scccccccummy - we need some viakale and to think there is anything wrong with us just shut up and think about it, there should be black people living there | "h" | True | "h" | "h" |
| Islamic Relief should be banned (not hate*) | Islamic Relief should be banned, so should islamic people (hate) | Islamic Relating To The Devil * | "h" | True | "h" | "h" |

Table 12: Selected examples from ICE(R2) attacks on the Hate Detection Task. This method generates text that learn from past seen text. Common failure modes include generating nonsensical sentences (see: row 5) or not being able to correctly label text (see: row 1). Some of the generated text are very random compared to the original examples. Often irrelevant text is added (see: row 7). Some of these lift phrases from other examples (ex: "deserve rights" in row 2 comes from another example). Sometimes the original label itself is incorrect (marked with *).

| Original Hate Speech R2a Example($a_0$) | Matched Hate Speech R2b Example($a_0$) | TextFooler(R2): Generated Example ($a_g$) | Desired Label | Label Correct? | Base(R1) ($\widehat{y}_{\theta_0}$) | Base(R1) + R2 ($\widehat{y}_{\theta_1}$) |
|---|---|---|---|---|---|---|
| are black folks extinct? cause they should be (hate) | are black panthers extinct? (not hate) | are black folks missing? cause they should be | "h" | True | "h" | "h" |
| lesbian r subhummmmman (hate) | Why the fuck do you think lesbians are subhuman? (not hate) | lesbian rs subhummmmman | "h" | True | "h" | "h" |
| Who would cut off their wiener for a hole...dont get it. it's like my brain just farts when I try to think of what must go through these people's heads to ever contemplate making that sort of crazy choice (hate) | Who would cut their wiener in several pieces for lunch..dont get it. it's like my brain just farts when I try to think of what must go through these people's heads to ever contemplate making that sort of crazy choice. (not hate) | Who would chop off their wiener for a hole...dont get it. it's like my brain just farts when I try to think of what must go through these people's heads to ever contemplate doing that sorted of psychopath choice | "h" | True | "n" | "h" |
| I can believe that she thinks arabs are the most kind ethnic group, like come on they are lovely (not hate) | I cant believe that she think arabs are the most kind ethnic group, like come on more like least (hate) | I can believe that she idea arabs are the most kind ethnic group, like come on they are lovely | "n" | True | "h" | "n" |
| black people are super clean (not hate) | negroes are not clean (hate) | negro people are super clean | "n" | True* | "h" | "n" |
| My work doesn't cater for people with invisible disabilities. (not hate) | Glad my work doesn't cater for people with disabilities. (hate) | My work doesn't cater for people with invisible disability. | "n" | True | "h" | "n" |
| look dude, they are evil and basically cant be trusted. Especially with their cheating crystal balls, they freak me the fuck out. Grubby looking witches. (not hate) | look dude, gyps are evil and basically cant be trusted. Especially with their cheating crystal balls, they freak me the fuck out. Grubby looking caravan lovers. (hate) | look dude, they are wicked and basically didnt be trusted. Especially with their cheating crystal balls, they freak me the fuck out. Grubby looking witches. | "n" | False | "n" | "n" |
| your personality is so ugly it could use makeup, typical black chick (hate) | your personality is so ugly it could use makeup (not hate) | your personality is so ugly it could use makeup, typical black dame | "h" | True | "n" | "h" |
| all women are not to be trusted, they are snakes, vicious vipers trying to trick their men (hate) | all men are not to be trusted, they are snakes, vicious vipers trying to trick their women (not hate*) | all men are not to be trusted, they are serpents, vicious vipers trying to trick their women, | "h" | False** | "h" | "n" |
| the bathroom is scummy we need some viakal and to get the cleaning sqaud in here. Maybe I'll do that tomorrow, ima go sort out somet stuff now (not hate) | the bathroom is so scummy it looks like an Arab house, honestly. We need some viakal and to get the cleaning sqaud in here. Maybe I'll do that tomorrow, ima go sort out somet stuff now (hate) | the toilettes is scummy we need some viakal and to get the cleaning sqaud in here. Possibly I'll do that tomorrow, ima go sort out somet stuff now | "n" | True | "n" | "n" |
| Islamic Relief should be banned, so should islamic people (hate) | Islamic Relief should be banned (not hate***) | Moslems Relief should be banned, so should islamic people | "h" | True | "h" | "h" |

Table 13: Selected examples from TextFooler(R2) attacks on the Hate Detection Task. As in the ANLI task, it makes random word substitutions. In this task, it emulates the style of creating the matched example pretty well, which is perhaps why we see AUC improvement within R2 in this task with this dataset (*Whether "negro" is pejorative is contextual, and this rating decision could go the other way. **We don't agree with this label, but it is consistent with the label choices in the dataset. ***This label comes from the dataset and may be incorrect.)

| Original Hate Speech R2a Example($\mathbf{a_0}$) | Matched Hate Speech R2b Example($\mathbf{a_0}$) | BertAttack(R2): Generated Example ($\mathbf{a_g}$) | Desired Label | Label Correct? | Base(R1) ($\widehat{y}_{\theta_0}$) | Base(R1) + R2 ($\widehat{y}_{\theta_1}$) |
|---|---|---|---|---|---|---|
| are black folks extinct? cause they should be (hate) | are black panthers extinct? (not hate) | are negro folks disappeared? reason they should be | "h" | True | "h" | "h" |
| lesbian r subhummmmmman (hate) | Why the fuck do you think lesbians are subhuman? (not hate) | lesbian rs subhummmmmman | "h" | True | "h" | "h" |
| Who would cut off their wiener for a hole...dont get it. it's like my brain just farts when I try to think of what must go through these people's heads to ever contemplate making that sort of crazy choice (hate) | Who would cut their wiener in several pieces for lunch..dont get it. it's like my brain just farts when I try to think of what must go through these people's heads to ever contemplate making that sort of crazy choice. (not hate) | Whom ought slice off their wiener for a hole...whats obtains it. it's likes my brain just farts when I endeavour to thinking of what owes going through these people's chiefs to ever envisage doing that sorts of maniac opted | "h" | True | "h" | "h" |
| I cant believe that she think arabs are the most kind ethnic group, like come on more like least (hate) | I can believe that she thinks arabs are the most kind ethnic group, like come on they are lovely (not hate) | I dunno believe that she believing arabs are the most types racism grouping, like arrive on more like fewer | "h" | ??? | "h" | "h" |
| black people are super clean (not hate) | negroes are not clean (hate) | nigger people are fabulous clean | "n" | False | "n" | "h" |
| My work doesn't cater for people with invisible disabilities. (not hate) | Glad my work doesn't cater for people with disabilities. (hate) | My corporation doesn't cater for people with undetectable incapacity. | "n" | True | "h" | "n" |
| look dude, they are evil and basically cant be trusted. Especially with their cheating crystal balls, they freak me the fuck out. Grubby looking witches. (not hate) | look dude, gyps are evil and basically cant be trusted. Especially with their cheating crystal balls, they freak me the fuck out. Grubby looking caravan lovers. (hate) | glance guy, they are wicked and primarily becuase be trustworthy. Specially with their cheating crystal balls, they flipper me the fucking out. Grubby looking bruges. | "n" | ???* | "n" | "n" |
| your personality is so ugly it could use makeup, typical black chick (hate) | your personality is so ugly it could use makeup (not hate) | your personality is so hideous it did utilise makeup, typical black dame | "h" | True | "n" | "h" |
| all women are not to be trusted, they are snakes, vicious vipers trying to trick their men (hate) | all men are not to be trusted, they are snakes, vicious vipers trying to trick their women (not hate*) | all men are not to be trustworthy, they are snakes, ferocious viper tempting to stratagem their mujer, | "h" | False** | "h" | "n" |
| the bathroom is scummy we need some viakal and to get the cleaning sqaud in here. Maybe I'll do that tomorrow, ima go sort out somet stuff now (not hate) | the bathroom is so scummy it looks like an Arab house, honestly. We need some viakal and to get the cleaning sqaud in here. Maybe I'll do that tomorrow, ima go sort out somet stuff now (hate) | the toilettes is scummy we needs some viakal and to obtain the cleansing sqaud in here. Likely I'll do that tomorrow, ima going sorted out somet thing now, | "n" | True | "n" | "n" |
| Islamic Relief should be banned, so should islamic people (hate) | Islamic Relief should be banned (not hate*) | Islamists Relief should be prohibiting, so should islamist people "h" | True | "h" | "h" | |

Table 14: Selected examples from BertAttack(R2) attacks on the Hate Detection Task. As in the ANLI task, it makes contextual word substitutions. In this task, it emulates the style of creating the matched example pretty well, which is perhaps why we see AUC improvement within R2 in this task with this dataset. Since it seeks out word synonyms from the model it attacks, it may have learned that "black people" and the slur, "nigger," are used to refer to the same category of people; it makes that substitution incorrectly in row 5 (*This label comes from the dataset and may be incorrect. **We don't agree with this label, but it is consistent with the label choices in the dataset.)

| Original Hate Speech R2a Example($\mathbf{a_0}$) | Matched Hate Speech R2b Example($\mathbf{a_0}$) | DI(R2): Generated Example ($\mathbf{a_g}$) | Desired Label | Label Correct? | Base(R1) ($\widehat{y}_{\theta_0}$) | Base(R1) + R2 ($\widehat{y}_{\theta_1}$) |
|---|---|---|---|---|---|---|
| are black folks extinct? cause they should be (hate) | are black panthers extinct? (not hate) | are black folks extinct? can anyone else now avoid living in our countryside with the extinction? | "h" | False | "h" | "h" |
| lesbian r subhummmmmman (hate) | Why the fuck do you think lesbians are subhuman? (not hate) | Why the fuck do you think lesbians are subhuman? | "h" | True | "n" | "n" |
| Who would cut off their wiener for a hole...dont get it. it's like my brain just farts when I try to think of what must go through these people's heads to ever contemplate making that sort of crazy choice (hate) | Who would cut their wiener in several pieces for lunch..dont get it. it's like my brain just farts when I try to think of what must go through these people's heads to ever contemplate making that sort of crazy choice. (not hate) | Who would cut their wiener in serious negroes for a hole...dont get it. it's like my brain just farts when I try to think of what must go through these people's heads to ever contemplate making that sort of crazy choice. | "h" | ???* | "n" | "h" |
| I cant believe that she think arabs are the most kind ethnic group, like come on more like least (hate) | I can believe that she thinks arabs are the most kind ethnic group, like come on they are lovely (not hate) | I can certainly believe that she thinks arabs are the most kind ethnic group, like come on they are lovely | "h" | False | "h" | "n" |
| black people are super clean (not hate) | negroes are not clean (hate) | black people are unclean | "n" | False | "h" | "n" |
| My work doesn't cater for people with invisible disabilities. (not hate) | Glad my work doesn't cater for people with disabilities. (hate) | Glad my work doesn't cater for people with invisible disabilities, always exempt from the amount of disabilities I can told me about their activities to show off other things | "n" | False | "h" | "n" |
| look dude, they are evil and basically cant be trusted. Especially with their cheating crystal balls, they freak me the fuck out. Grubby looking witches. (not hate) | look dude, gyps are evil and basically cant be trusted. Especially with their cheating crystal balls, they freak me the fuck out. Grubby looking caravan lovers. (hate) | they are evil and basically cant be trusted. Especially with their cheating crystal balls, they freak me the fuck out. Grubby looking with their cheating crystal balls. | "n" | ???* | "n" | "n" |
| your personality is so ugly it could use makeup, typical black chick (hate) | your personality is so ugly it could use makeup (not hate) | your personality is so ugly it could use makeup, typycal women do not deserve to be allowed to work with yellow neighbours | "h" | True | "h" | "h" |
| all women are not to be trusted, they are snakes, vicious vipers trying to trick their men (hate) | all men are not to be trusted, they are snakes, vicious vipers trying to trick their women (not hate**) | all black women are not to be trusted, they are serpents, vicious vipers trying to trick their women, | "h" | True | "h" | "h" |
| the bathroom is scummy we need some viakal and to get the cleaning sqaud in here. Maybe I'll do that tomorrow, ima go sort out somet stuff now (not hate) | the bathroom is so scummy it looks like an Arab house, honestly. We need some viakal and to get the cleaning sqaud in here. Maybe I'll do that tomorrow, ima go sort out somet stuff now (hate) | the bathroom is so scummy we need some viakal and to get the cleaning sqaud in here. Maybe I'll do that tomorrow, ima go sort out somet stuff now | "n" | True | "n" | "n" |
| Islamic Relief should be banned, so should islamic people (hate) | Islamic Relief should be banned (not hate**) | Islamic Relief is trash. We need Muslim like the retarded | "h" | True | "h" | "h" |

Table 15: Selected examples from DI(R2) attacks on the Hate Detection Task. This method on this task is trained with pairs that have opposite labels (one pair hate, the other not hate). As a result, the model often generates text closer to the matched example than to the original (ex: rows 3 to 5) which results in very high label confusion (*Label is unclear because while a group is not explicitly referred to, "crystal ball" can be a dog whistle. **This label comes from the dataset and may be incorrect.)

| No. of generated examples | ANLI R1 | ANLI R2 | ANLI R3 |
|---|---|---|---|
| 1k | 36.5 | 31.6 | 36.2 |
| 10k | 41.8 | 32.3 | 36.3 |
| 100k | 42.9 | 36.5 | 37.0 |
| 200k | 45.0 | 36.6 | 37.2 |
| 400k | 45.1 | 36.9 | 37.4 |
| 850k | **48.2** | **39.1** | **40.1** |
| 1.6M | 43.9 | 37.9 | 38.7 |

Table 16: Impact of the number of generated adversarial examples from DI(R1) used for the fine-tuning the base classifier, on future rounds of attack in ANLI R2 and R3

| Model | $a_0$: R1 | $a_1$: R2 | $a_2$: R3 |
|---|---|---|---|
| $\widehat{y}_{\theta_2}$: Base + R1 + R2 | 52.7 | 41.6 | 38.4 |
| $\hookrightarrow$ + DI (R1 + R2) | 52.8 | 44.2 | 42.3 |
| $\hookrightarrow$ + ICE(R1 + R2) | **55.2** | **48.1** | **42.8** |

Table 17: Improvement on ANLI accuracy (%) when trained on attacks generated only from Rounds 1 and 2. The notation, DI(R1 + R2), refers to the method DI using R1 and R2 data to generate more examples.

| step # | base+R1 | DI(R1) |
|---|---|---|
| 6,135 | **40.8%** | - |
| 12,270 | 37.5% | 36.4% |
| 18,405 | 33.3% | **42.2%** |
| 24,540 | 37.0% | 33.3% |
| 30,675 | 38.2% | 33.3% |
| 36,810 | 38.5% | 37.3% |
| 42,945 | - | 41.5% |

Table 18: Comparison of ANLI R1 validation set accuracy when training on additional steps on ANLI R1 data, as compared to using the best checkpoint from ANLI R1 data, and using that to further train on generated DI(R1) adversarial examples.

| Model | ANLI R1 | SNLI | MNLI(m) |
|---|---|---|---|
| Base | 0.21 | 0.90 | 0.86 |
| Base + R1 | 0.44 | 0.82 | 0.77 |
| Base + R1 + TextFooler(R1) | 0.24 | 0.13 | 0.07 |
| Base + R1 + DI(R1) | 0.48 | 0.74 | 0.69 |
| Base + R1 + ICE(R1) | 0.51 | 0.75 | 0.66 |

Table 19: Trade-off between adversarial and clean accuracy as measured on ANLI R1 and MNLI/SNLI test data

| Dataset | Round | Train | Validation | Test |
|---|---|---|---|---|
| | 1 | 16946 | 1000 | 1000 |
| ANLI | 2 | 45460 | 1000 | 1000 |
| | 3 | 100459 | 1200 | 1200 |
| | 1 | 8844 | 1091 | 1111 |
| | 2 | 7998 | 999 | 999 |
| HateSpeech | 3 | 7960 | 995 | 995 |
| | 4 | 8122 | 1015 | 1015 |

Table 20: Number of human adversarial examples in ANLI and Hate speech detection task, by split.

| Accuracy on Test Set Round → Pearson correlation (p-value) ↓ | R1 | R2 | R3 |
|---|---|---|---|
| Attack success rate | -0.98 (0.20) | -0.76 (0.24) | -0.57 (0.43) |
| Label noise | 0.28 (0.82) | 0.47 (0.69) | 0.19 (0.88) |
| Distributional similarity (MAUVE) | 0.28 (0.55) | 0.03 (0.95) | -0.18 (0.71) |

Table 21: Pearson correlation between accuracy on test sets of ANLI across the three rounds with the metrics of the datasets that were augmented to obtain the test set accuracies. p-values in braces show that the correlation coefficients are not statistically significant if we were to consider a significance level of 0.05.

| | Original dataset (ANLI R1) | | Generated dataset (ANLI R1) | |
|---|---|---|---|---|
| Rank | trigram | Count | trigram | Count |
| 1 | [PAD] [PAD] [PAD] | 88702721 | [PAD] [PAD] [PAD] | 17559312 |
| 2 | [SEP] [PAD] [PAD] | 940829 | [SEP] [PAD] [PAD] | 681634 |
| 3 | . [SEP] [PAD] | 803393 | . [SEP] [PAD] | 551759 |
| 4 | . [SEP] the | 225175 | . [SEP] the | 167479 |
| 5 | . [SEP] a | 215833 | ) is a | 153471 |
| 6 | [CLS] a man | 88185 | . it is | 147691 |
| 7 | [SEP] a man | 60532 | . it was | 108310 |
| 8 | a man is | 41999 | ) is an | 86666 |
| 9 | [CLS] a woman | 38948 | is an american | 84950 |
| 10 | . [SEP] there | 37004 | . [SEP] | 60869 |
| 11 | . [SEP] two | 35716 | . the film | 59364 |
| 12 | a group of | 34385 | the united states | 59019 |
| 13 | a man in | 34221 | one of the | 55591 |
| 14 | in front of | 33005 | . he is | 50200 |
| 15 | it ' s | 32399 | . he was | 49679 |
| 16 | [SEP] the man | 32013 | film directed by | 48994 |
| 17 | man in a | 30826 | ) was an | 43507 |
| 18 | [SEP] a woman | 28811 | . s . | 41190 |
| 19 | don ' t | 26569 | best known for | 41149 |
| 20 | [CLS] a young | 24398 | it is the | 39018 |
| 21 | the man is | 23170 | and | 38158 |
| 22 | front of a | 21500 | u . s | 37818 |
| 23 | [CLS] a group | 19751 | member of the | 34558 |
| 24 | a woman is | 19629 | based on the | 33319 |
| 25 | [SEP] there is | 18052 | united states . | 32740 |
| 26 | street . [SEP] | 17960 | , | 32723 |
| 27 | ##s . [SEP] | 17820 | in the united | 32610 |
| 28 | group of people | 17633 | was an american | 30761 |
| 29 | that ' s | 17395 | ) was a | 30612 |
| 30 | [SEP] there are | 17380 | , and the | 30556 |
| 31 | . [SEP] people | 16961 | ) is | 30225 |
| 32 | a woman in | 16732 | a member of | 30158 |
| 33 | outside . [SEP] | 16731 | known for his | 29994 |
| 34 | there is a | 15516 | of the same | 29508 |
| 35 | woman in a | 15426 | is a | 29066 |
| 36 | water . [SEP] | 15068 | as well as | 29046 |
| 37 | [SEP] the woman | 14418 | , is a | 28809 |
| 38 | it . [SEP] | 14193 | ) . [SEP] | 28183 |
| 39 | [CLS] two men | 12992 | the film was | 28100 |
| 40 | i ' m | 12961 | was the first | 27252 |
| 41 | i don ' | 12864 | ##hh ##hh ##hh | 26073 |
| 42 | . [SEP] he | 12109 | was released in | 26007 |
| 43 | [SEP] a person | 11867 | also known as | 25945 |
| 44 | . [SEP] it | 11531 | the same name | 25307 |
| 45 | in the background | 11402 | part of the | 25297 |
| 46 | [SEP] a group | 11329 | . the album | 25120 |
| 47 | sitting on a | 11171 | it was released | 24839 |
| 48 | . [SEP] they | 11052 | known as the | 24186 |
| 49 | a man with | 10863 | and directed by | 24126 |
| 50 | a man and | 10827 | is one of | 24118 |
| 51 | a man wearing | 10723 | , united states | 23211 |
| 52 | next to a | 10662 | the u . | 22836 |
| 53 | the woman is | 10650 | is the first | 22140 |
| 54 | in a blue | 10484 | is a song | 22096 |
| 55 | . [SEP] i | 10446 | , england . | 21857 |
| 56 | [SEP] a boy | 10085 | s " | 21520 |
| 57 | beach . [SEP] | 10071 | it is a | 21373 |
| 58 | [SEP] the people | 10021 | . the song | 21146 |
| 59 | the street . | 9928 | . he has | 20997 |
| 60 | [SEP] a girl | 9877 | the university of | 20995 |
| 61 | [SEP] people are | 9870 | , " the | 20964 |
| 62 | background . [SEP] | 9721 | the population of | 20000 |
| 63 | didn ' t | 9702 | was released on | 19948 |
| 64 | [CLS] a person | 9550 | it was the | 19866 |
| 65 | building . [SEP] | 9528 | he was the | 19396 |
| 66 | a lot of | 9472 | , and is | 19382 |
| 67 | him . [SEP] | 9465 | the civil parish | 18494 |
| 68 | [CLS] a little | 9404 | was born in | 18125 |
| 69 | man wearing a | 9313 | located in the | 18112 |
| 70 | ? [SEP] [PAD] | 9301 | . the | 18087 |
| 71 | [CLS] a boy | 9264 | ) , " | 17984 |
| 72 | them . [SEP] | 9037 | is based on | 17797 |
| 73 | a young boy | 8945 | , and was | 17420 |
| 74 | [CLS] a girl | 8924 | is best known | 17371 |
| 75 | . [SEP] an | 8875 | , and | 16806 |
| 76 | the background . | 8844 | , was a | 16718 |
| 77 | in a red | 8771 | , it was | 16390 |
| 78 | man with a | 8720 | , is an | 16383 |
| 79 | in a white | 8646 | album , " | 16325 |
| 80 | is wearing a | 8598 | the 2010 census | 16280 |
| 81 | the people are | 8552 | , in the | 16136 |
| 82 | two men are | 8099 | written and directed | 16058 |
| 83 | in a black | 8084 | is the second | 15720 |
| 84 | . [SEP] three | 7996 | the 2011 census | 15671 |
| 85 | [SEP] two men | 7963 | . the population | 15631 |
| 86 | they ' re | 7962 | is located in | 15591 |
| 87 | . [SEP] some | 7881 | the city of | 15439 |
| 88 | [SEP] a dog | 7866 | . she was | 15402 |
| 89 | field . [SEP] | 7758 | drama film directed | 15052 |
| 90 | [SEP] a young | 7747 | is a former | 14879 |
| 91 | park . [SEP] | 7741 | s first | 14875 |
| 92 | a person is | 7732 | men ' s | 14871 |
| 93 | the water . | 7539 | , also known | 14838 |
| 94 | ball . [SEP] | 7535 | population of the | 14758 |
| 95 | a little girl | 7350 | ) . it | 14443 |
| 96 | a young man | 7345 | women ' s | 14232 |
| 97 | of people are | 7330 | . she is | 14207 |
| 98 | i ' ve | 7307 | a song by | 14039 |
| 99 | a young girl | 7236 | the town of | 13895 |
| 100 | s a | 7179 | ) , and | 13840 |

Table 22: 100 most-common trigrams in original and DI-generated dataset for ANLI R1 dataset

