# OpenReview forum: "Break it, Imitate it, Fix it: Robustness by Generating Human-Like Attacks"
_TMLR — Accepted by TMLR_

### Review · Reviewer_1LU8 · 2023-11-13

**Summary Of Contributions:**

The paper introduces a novel adversarial training framework for natural language processing (NLP) systems. It focuses on enhancing the robustness of these systems against human adversaries, a crucial aspect given the increasing sophistication of adversarial attacks. The primary contributions and new knowledge presented by this submission are as follows:

1. The paper proposes an adversarial training framework that uses a limited number of human adversarial examples to generate additional synthetic, human-like adversarial examples at scale. This method contrasts with traditional approaches that either directly train on real adversarial examples or rely on simpler synthetic attacks.

2. The framework is tested on benchmark datasets like ANLI and hate speech detection. The results demonstrate significant improvements in model robustness against future rounds of human-generated attacks.

**Audience:**

Yes

**Broader Impact Concerns:**

Ethical Implications Not Sufficiently Addressed

1. The paper's framework enhances the robustness of NLP systems against adversarial attacks. However, there is a risk that malicious actors could adapt their strategies in response to these advancements.

2. The risk of overfitting to specific types of adversarial attacks, particularly those generated based on a limited dataset, could inadvertently introduce or reinforce biases in the model.

3. Improved generative text modeling could make it more challenging for humans to distinguish between human-generated and machine-generated text.

4. The development of more robust NLP systems might disproportionately benefit entities with greater resources, potentially widening the gap between well-funded organizations and smaller entities.

**Claims And Evidence:**

No

**Requested Changes:**

1. **Clarify Technical Notation and Concepts (Section 3)**:
   - *Description*: The paper should provide clearer definitions and distinctions between key concepts, such as the base classifier and the new classifier.
   - *Rationale*: This clarity is essential for understanding the fundamental aspects of the proposed framework.
   - *Impact*: Without this clarification, the paper risks misinterpretation or misunderstanding of its core methodologies.

2.  The language in Section 4 should be simplified, reducing the reliance on excessive symbols and technical jargon.


3. Redesign or simplify Figure 1 to make the workflow of the proposed framework more intuitive.

4. Include evaluations in scenarios with fewer human adversarial examples and across a broader range of classifiers.

**Strengths And Weaknesses:**

Strengths
1. The submission introduces a novel framework for adversarial training, effectively utilizing limited human adversarial examples to generate synthetic attacks.

2. The study empirically demonstrates that existing synthetic attack methods do not effectively improve robustness against real human-generated attacks.


Weakness
1. One notable weakness in the submission pertains to the clarity in the presentation of technical content, particularly in Sections 3 and 4, as well as in the accompanying figures.

    - Ambiguity in Notation and Concepts (Section 3): The paper introduces various notations and concepts, such as the base classifier and the new classifier, without clear definitions or distinctions. This lack of clarity hinders the reader's understanding of the fundamental aspects of the proposed framework, such as how the new classifier differs from or improves upon the base classifier. Clearer definitions and possibly a simplified explanation or illustration of these concepts would greatly enhance comprehension.

    - Complexity in Technical Writing (Section 4): The writing in Section 4, which details the overall solution framework, suffers from an overabundance of symbols and technical jargon. This complexity makes it challenging for readers to grasp the core ideas and the logical flow of the proposed methods. Simplifying the language and reducing the reliance on symbols could make the section more accessible to a broader audience.

    - Complexity in Visual Representation (Fig. 1): The workflow illustrated in Figure 1 is described as overly complex, potentially confusing readers. A more streamlined and intuitive visual representation could significantly aid in conveying the framework's workflow more effectively.

2. The paper acknowledges that if more robust models are employed, human adversaries might adapt to them. This aspect was not investigated, and it remains uncertain whether the new models would still be resilient against these adapted attacks.

3. The effectiveness of the methods was tested with a substantial number of human adversarial examples but not in scenarios with extremely few examples. Additionally, the study did not evaluate the methods on classifiers with access to larger real adversarial datasets or diverse model architectures.

4. There is a concern that the new classifiers trained with generated data might overfit to the human-generated adversarial data, potentially leading to lower performance on different future attacks or original tasks.

---

> ### Author Response · Authors · 2023-12-20
> **Response to review**
>
> We appreciate the reviewer's insightful feedback and have addressed them in order.
>
> > Clarity in Sections 3 and 4
>
> We have revised these sections for improved clarity (see pages 3-5 and Alg 1 in Sec 4).
>
> > Adaptability of human adversaries
>
> We acknowledge the potential for human adversaries to adapt, and acknowledge the lack of access to live human attackers is a limitation of our work. This limitation is discussed in both the “Limitations” and “Overall solution framework” sections. The ICE and DI methods nevertheless lead to robustness to attacks on which the Control (base model trained only on round 1 attacks) is not robust to. We retrained the Control 3 times to ensure this increased robustness was not a one-off result.
>
> > Few-shot generalization
>
> The reviewer notes that we do not test scenarios with extremely few examples. Table 6 in the paper does test our methods on as few as 100 examples, and notes that our amplification methods lead to increased robustness with as few as 500 examples provided. As we were not able to improve in the 100 example setting, we did not test with even fewer examples. Moreover, as outlined in Section 6C, in the extremely few example settings (n ~ 10), the relevant baseline methods are different as well, as we increasingly need to be concerned about attack coverage: whether we have used a diverse set of attacks to begin with. Without such a coverage analysis in the n ~ 10 scenario, the scope of our problem formulation would be significantly expanded to achieve out-of-distribution (OOD) generalization. We do not claim to improve OOD generalization in this paper. Instead, we rely on existing human adversarial attack datasets, and aim to be robust to attacks from human adversaries in future rounds of attacks. Categorizing attack subtypes and partitioning the dataset based on these subtypes is non-trivial and beyond the scope of our work.
>
> > Efficacy under large data availability scenarios
>
> The reviewer also notes the counterpart: asking whether large enough real adversarial datasets might negate the need for data amplification. We agree and share their concern that enough data might render amplification unnecessary. However, obtaining large, diverse adversarial datasets for LLMs is incredibly expensive (Xu et al., 2021; Hendrycks et al., 2021). We utilized the largest available dataset (featuring iterative attacks on an evolving model) and have shown improvement in robustness. We welcome suggestions for larger datasets if available.
>
> > Different model architectures
>
> We chose BERT-Large as the classifier (as this is the model type that was attacked in the DynaBench datasets we use). And then we used T5 as the matching generator for two reasons: 1.) It matched the size and capabilities of BERT-Large on these tasks, while running on a very different architecture and training set-up. This controlled for potential improvements due to model size differences. 2. The transformer Encoder-Decoder is now the ubiquitous generative model architecture. Improvements to model architecture are beyond the scope of this paper which focuses on data augmentation narrowly as outlined in the Sec 2: related work. We have added this critique to our limitations section.
>
> > Robustness vs accuracy trade-offs
>
>
> The reviewer astutely points out a potential trade-off: increasing robustness to human-generated attacks might hurt performance on other data distributions (ex: models trained on our synthetic examples likely perform poorly on word substitution attacks, whereas a model trained on TextFooler attacks will perform well there). This concern is precisely what sparked our research: how to boost adversarial robustness within the constraints of the model's size and capability, especially when a larger, "outperforms-all" model isn't readily available. Hence, our paper asks: how can we improve robustness against tricky examples while staying within the limitations of the base model's size and capabilities?
>
> > Ethical Concerns:
>
> We appreciate the reviewer's thorough analysis of ethical concerns. We share the top three concerns, which is why they are now highlighted in our limitations section. Regarding the final point, however, we believe our work empowers users to refine their models for specific data distributions without relying on expensive, resource-intensive "bigger-better" models. This opens doors for medium-sized entities (those capable of fine-tuning T5 models) to compete with well-funded organizations by tailoring their smaller model’s robustness to their specific context.

---

> ### Comment · Reviewer_1LU8 · 2023-12-25
> **Response to authors**
>
> The concerns are well-resolved.

---

### Review · Reviewer_jMpW · 2023-11-18

**Summary Of Contributions:**

This paper focuses on improving the robustness of natural language processing (NLP) systems against adversarial attacks by using limited human adversarial samples. It first presents a study that demonstrates the existing synthetic attack approaches often fail to improve robustness against real human adversarial attacks. Then the authors propose a framework that uses generative models to imitate real human-generated attacks, which enhances robustness. Overall, the paper contributes to the field of adversarial robustness in NLP by proposing a novel approach that leverages the imitation of real human attacks, questioning the effectiveness of commonly used metrics in the domain, and demonstrating the practicality of its methods through empirical evaluation.

**Audience:**

Yes

**Claims And Evidence:**

Yes

**Requested Changes:**

Critical:
* Concerning Potential Bias Issues:
The approach adopted in the 'Warm-starting' technique may introduce bias in the adversarial examples it generates.
This potential issue arises from the method's preference for text phrases with pre-established labels, leading to the reconstruction of previously observed attacks. While this approach simplifies the task for the classifier, it risks creating a limited scope of adversarial examples that might not fully represent the variety of real-world adversarial inputs.
These biases could skew the model's learning towards certain language patterns or themes, potentially affecting its ability to handle diverse adversarial attacks. A more thorough discussion of how the methodology accounts for and mitigates this bias would greatly enhance the paper.

* On the Insufficiency in Result Explanation:
The paper mentions that commonly used metrics are not correlated with actual model performance, yet there seems to be no calculation of Pearson Correlation Coefficient or similar statistical measures to substantiate this claim.
While the paper lists various tables showing experimental results, like Tables 4 and 5, there is a notable lack of in-depth analysis accompanying these tables. The data are described, but there is little to no discussion on why certain anomalies might be occurring. This paper could benefit from more robust empirical evidence or a deeper theoretical analysis to validate the effectiveness of their proposed methods over these traditional metrics

Minor  comments:
* Section 4 of the paper, titled "Synthetic Attack Generators," mentioned that “We now present two methods..”. However, there are four bullet points listed below: It may cause confusion.

* In the 'Warm-Starting' subsection, there appears to be a disconnect between the second and third paragraphs. The transition between these paragraphs is not seamless, creating a potential gap in the logical flow of ideas.

**Strengths And Weaknesses:**

Strengths:
* The paper targets a critical issue in its field of study.
* This paper is easy to understand. Its workflow diagram effectively illustrates the functionality of the framework.
* The Results section presents experimental data in detail.

Weaknesses:
* Potential for bias in the generated adversarial examples due to this preference for certain words or phrases in reconstructions.
* Insufficient explanation in Results section.

---

> ### Author Response · Authors · 2023-12-20
> **Response to review**
>
> We thank the reviewer for their thoughtful and actionable feedback on our work.
>
> > Concerning Potential Bias Issues
>
> While the reviewer rightly points out our method (ICE’s) may have a bias towards previously seen attacks that potentially limits attack diversity (shown in Appendix Table 11, and by the high distributional similarity in Fig. 2), we argue that learning the distribution of the past attacks can still be beneficial. Recent work (“Data Distributional Properties Drive Emergent In-Context Learning in Transformers” - https://arxiv.org/abs/2205.05055) suggests LLMs learn better when the same concepts are repeated, especially when seen in new contexts; ICE facilitates exactly this. This repetition in slightly varied contexts may help the model learn the essence of the original human attacks, leading to improved performance against similar attacks. We acknowledge, however, that our methods do not guarantee robustness to unseen attack types, nor does it improve overall model performance.
> Further, to analyze if our methods rely on identifying phrases of text that might be adversarial in nature without imitating human adversarial text patterns, we analyzed the top 100 trigrams in both the original and generated datasets. In Table 22 of the Appendix, we found no such anomalies to indicate that universal adversarial patterns might be getting generated and thus generating potential bias at a token level. However, we acknowledge the risks and negative consequences related to the generated text in the limitations section (Pu et al., 2023) and the release of such adversarial data should be carefully considered.
> > On the Insufficiency in Result Explanation
>
> We acknowledge the absence of a theoretical or empirical mechanistic rationale for why the common heuristics (listed below) are ineffective at predicting future robustness (We posit our current hypothesis for why the ICE method works in Sec 6D). Moreover, we have calculated the Pearson correlation coefficient, as suggested, to further support that there is no significant correlation between each of the dataset heuristics and robustness to future rounds of attacks: In Table 21, we see that none of the Pearson correlation coefficients are significant (p-value > 0.05). We refer to this analysis in the main text (Sec 6B).
>
> Defining a new data quality metric that is successful at predicting robustness to future attacks is beyond the scope of our work. This is because validating such a new metric (without overfitting to exactly what worked in our current methods and experiments) would be non-trivial.
>
> ANLI Round # → Heuristic ↓ | R1            | R2            | R3            |
> | -------------------------------------------------------- | ------------- | ------------- | ------------- |
> | Attack success rate                                      | \-0.98 (0.20) | \-0.76 (0.24) | \-0.57 (0.43) |
> | Label noise                                              | 0.28 (0.82)   | 0.47 (0.69)   | 0.19 (0.88)   |
> | Distributional similarity (MAUVE)                        | 0.28 (0.55)   | 0.03 (0.95)   | \-0.18 (0.71) |
>
> P-values in parenthesis (<0.05 typically considered significant, not considering multiple tests)
>
> > Minor comments
>
> We have incorporated the writing feedback to improve readability of Sections 3 and 4 of the paper.

---

### Review · Reviewer_SM36 · 2023-11-22

**Summary Of Contributions:**

The authors propose a method to imitate human-generated adversarial examples in NLP tasks. They show that augmenting the training data with extra adversarial examples modeled on human-generated adversarial examples can improve robustness to further adversarial attacks from humans. They also provide a study on different properties of adversarial examples generated by different methods, and correlate their usefulness with other properties like distributional similarities and label noise level. They show that their method is more robust against human attacks on the ANLI and Hate Speech Detection datasets.

**Audience:**

Yes

**Claims And Evidence:**

Yes

**Requested Changes:**

- Given the importance of the proposed DI and ICE adversarial example generation methods, the author should consider moving the algorithm description from the appendix to the main text of the paper. There is space available as a journal paper.

- Clean up the algorithm description, use proper and consistent notations as commonly used in the literature.

- There are duplicate references in Ganguli et al 2022, Jia et al 2019, Raffel et al 2020. Please fix.

**Strengths And Weaknesses:**

Strengths:
- The results of this paper are very interesting. It studies imitating human generated adversarial examples to improve the robustness of neural language models. It shows that models trained on the first round of human-generated adversarial examples and their imitations are more robust towards subsequent human-generated attacks in round 2 and round 3.

- This study also provides many interesting conclusions from analyzing the generated adversarial examples by different methods and their relation to robustness. For example, it finds that distributional similarities and level of label noise does not directly predict the usefulness of training with those adversarial examples.

- The paper is well-structured and easy to follow. It is quite clear except for the section related to algorithm descriptions.


Weaknesses:
- The empirical evaluation is done only on two datasets, Adversarial NLI and Hate Speech Detection, which is somewhat limited.

- On p4, the authors claim that the learned adversarial example distribution \hat{p_a0} is similar to the future adversarial distributions p_a1, p_a2. This statement seems unsupported because you cannot draw this conclusion by simply observing that training on \hat{p_a0} helps with defending against p_a1 and p_a2. Using analogy from computer vision, although a model trained on FGSM (one-step gradient) is more robust against multi-step (e.g. 5 or 10) PGD attacks than a clean model, the adversarial examples generated from these methods are very different.

- The algorithm description can be improved to make it more self-contained. When describing the synthetic attack generators DI and ICE used in this paper, the authors refer frequently to existing methods like T5 and Plug and Play controlled decoding methods, which is fine in itself. But it would be better to describe the ideas directly without having the readers having to look up those papers separately. Also having a self-contained description of the method is also important since it makes the modifications employed by the authors easier to understand. Currently it is not easy to re-implement what the authors describe due to lack of details.

- The notations used in the algorithm description also makes the paper difficult to understand. In Algorithm 1, methods like G.train and G.loss G.backpropagate are not defined. It would be better to define a loss function for G (the loss should be independent from G itself), or just say do backprogagation on G instead of using these confusing notations. In the inner loop the computation of GRAD and G'.backpropagate is confusing because gradient computation is done via backpropagation and they should be the same thing. This part is very confusing and needs to be rewritten.

---

> ### Author Response · Authors · 2023-12-20
> **Response to review**
>
> We thank the reviewer for their thorough and careful review of our paper and are glad they find our results interesting.
>
> > algorithmic clarity
>
> The updated explanation in the main paper (pg 5-6, and Alg 1 in Sec 4) should now be self-sufficient for understanding the algorithms. Additionally, we adopted the reviewer's notation suggestions by describing the training inputs and losses in Alg 1 (Sec 4) more clearly with additional details in the appendix.
> > limited empirical results only on two datasets
>
> We test on two datasets, as these are the two datasets we found that that meet our criteria. Specifically, open-source datasets with iterative human-and-model-in-the-loop attacks are limited; only 4 of the 24 challenge datasets in Dynabench have more than 1 round of data. The other two datasets were excluded because the training data was not available--namely German hate speech dataset and Question Answering (Only the challenge datasets are available which does not enable us to compare models in our set up). The two datasets we did use represent diverse tasks: entailment (long attacks, many per prompt) and hate speech (short attacks, few per prompt). We welcome suggestions for additional datasets with varied attack settings.
> > distributional generalization
>
> We agree that our claim about \hat{p_a0} resembling future adversarial distributions is unsupported by the experiments (And have removed the original claim, and stated the new caveat in Section 4, Overall Solution Framework sub-section). While our empirical results on generalizing to new rounds of adversarial attack hold promise, the connection between rounds of adversarial attacks should be further studied in future work.
>
> > references
>
> Thanks to the reviewer, we've corrected duplicate references in Ganguli et al 2022, Jia et al 2019, and Raffel et al 2020.

---

### Author Response · Authors · 2023-12-20
**Summary of response to reviewers**

We thank all the reviewers for their valuable feedback, and the AE for facilitating them. We believe that the feedback helped us improve the readability and soundness of the paper. We address each of the reviews with detailed point-by-point replies and we hope this further clarifies our contributions in the paper. Broadly, the revision consists of the following changes (highlighted in red in the revised PDF):
* Rewrote methods and algorithm sections to improve clarity
* Added correlation analysis to show that existing measures are not significantly correlated with future human adversarial robustness
* Addressed concerns around potential bias through qualitative analysis of generated text
* Expanded on the limitations and risks sections
* Fixed references and typos

---

### Decision · Action_Editor_c5Yb · 2023-12-29

**Recommendation:** Accept with minor revision

**Comment:**

This paper proposes using (limited) human feedback for adversarial training of language models. All reviewers and the editor find the proposed method novel and the results convincing. The authors's rebuttal has successfully addressed most of the reviewers' concerns. However, the authors are suggested to do the following necessary revisions in the final version:

- This paper only used two datasets for evaluation. Can the authors include the results from a new dataset?
- Please address the comment on the similarity between single-step and multi-step adversarial example distributions raised by Reviewer SM36, instead of treating it as future work.

I believe these two revisions will greatly improve the quality and impact of this work.

**Audience:**

This topic (AI robustness) fits into the general audience of AI / machine learning

**Claims And Evidence:**

The experimental results (while limited to two datasets) support the claims and empirical evidence. The paper is well written and explains the findings clearly.

---

> ### Author Response · Authors · 2024-01-26
> **Revision**
>
> We thank the editor and reviewers for helping us improve the paper throughout the feedback period. We look forward to obtaining any further clarifications on the revised paper, based on our response below.
>
> > only two datasets for evaluation
>
> We acknowledge that our method has been evaluated on 2 datasets. However, as noted in our response to Reviewer SM36, this was due to a lack of publicly available adversarial datasets with iterative human-and-model-in-the-loop attacks against a model that is at least as capable as Bert Large. We referred to the latest version of Dynabench, where only 4 of the 24 challenge datasets in Dynabench have more than 1 round of data. The other two datasets were excluded because the training data was not available--namely German hate speech dataset and Question Answering (Only the challenge datasets are available which does not enable us to compare models in our set up). We welcome suggestions for alternative datasets that meet this criteria.
>
> > similarity between single-step and multi-step adversarial example distributions
>
> We do have MAUVE metrics that compare the distributional similarity of the synthetic augmented attacks and Future real human attacks (See Fig. 2). Though MAUVE is a SOTA distributional similarity metric for text, no clear pattern emerges: MAUVE is not able to make clear conclusions about the relationship between which synthetic attacks will most help be robust to future human attacks (It is not the case that the highest distributional similarity synthetic attack is best -- TextFooler). We have expanded on this in the discussion of the results (Section B.I).
>
> Reviewer SM36 makes an analogy to vision adversarial work where although training on FGSM single-step attacks makes a model more robust to multi-step attacks, that does not mean the single step attack distribution and the multi-step attack distribution are similar.
> We agree with this analogy and Reviewer SM36’s point, and have modified our paper (See: Sec 4: Overall Solution Framework sub-section at the top of page 5) to clearly reflect that: Namely, although training on our augmented attacks makes a model more robust to future human attacks, that does not mean the augmented attacks and the future human attack distribution are the same distribution. In doing so, we address their comment.
>
> The precise nature of the relationship between the two distributions in each pair is not well understood (Pair 1: Vision single-step FGSM attack : Vision multi-step attack; Pair 2: synthetic augmented attacks : Future real human attacks).  That is to say, better understanding when training on one distribution will generalize to another is an open problem in the literature (applies to both problems).

---

> > ### Comment · Action_Editor_c5Yb · 2024-01-29
> > **Accept the revised version**
> >
> > I thank the authors for the detailed responses and updates based on the suggested changes. I've reviewed the updated version and found the changes satisfactory. Therefore, I approve the acceptance of this paper. Congratulations again to the authors, and I thank the reviewers for their valuable comments.
> >
> > Action Editor